# Extreme Wind Shear Events in U.S. Offshore Wind Energy Areas and the Role of Induced Stratification

Mithu Debnath[1], Paula Doubrawa[1], Mike Optis[1], Patrick Hawbecker[2], and Nicola Bodini[1]

[1]National Renewable Energy Laboratory, Golden, Colorado, USA
[2]National Center for Atmospheric Research, Boulder, Colorado, USA

**Correspondence:** Mithu Debnath (mithu.debnath@nrel.gov)

**Abstract.** As the offshore wind industry emerges on the U.S. East Coast, a comprehensive understanding of the wind resource—particularly extreme events—is vital to the industry's success. Such understanding has been hindered by a lack of publicly available wind profile observations in offshore wind energy areas. However, the New York State Energy Research and Development Authority recently funded the deployment of two floating lidars within two current lease areas off the coast of New Jersey. These floating lidars provide publicly available wind speed data from 20-m to 200-m height with a 20-m vertical resolution. In this study, we leverage a year of these lidar data to quantify and characterize the frequent occurrence of high wind shear and low-level jet events, both of which will have a considerable impact on turbine operation. In designing a detection algorithm for these events, we find that the typical, non-dimensional power law-based wind shear exponent is insufficient to identify many of these extreme, high wind-speed events. Rather, we find that the simple vertical gradient of wind speed better captures the events. Based on this detection method, we find that almost 100 independent events occur throughout the year with mean wind speed at 100-m height and wind speed gradient of 16 m s$^{-1}$ and 0.05 s$^{-1}$, respectively. The events have strong seasonal variability, with the highest number of events in summer and the lowest in winter. A detailed analysis reveals that these events are enabled by an induced stable stratification when warmer air from the south flows over the colder mid-Atlantic waters, leading to a positive air-sea temperature difference.

## 1 Introduction

The offshore wind industry is rapidly developing on the U.S. East Coast and a comprehensive understanding of the wind resource in this area is critical for the industry's success. There are currently 15 active lease areas with over 21 Gigawatts (GW) of planned capacity spanning from Massachusetts to North Carolina (Fig. 1), with an additional planned 86-GW capacity in all

U.S. waters by 2050 (BOEM, 2018). Proposed lease areas are located on the Atlantic Outer Continental Shelf (OCS) and span locations ranging from a minimum of 15 km to a maximum of over 100 km from the coastline. The proper planning, design, and operation of these wind farms require an in-depth understanding of the wind characteristics in the OCS, in particular the frequency and magnitude of extreme events that largely impact the power performance, safety, and operation of wind turbines (Musial and Ram, 2010; Rose et al., 2012; Archer et al., 2014).

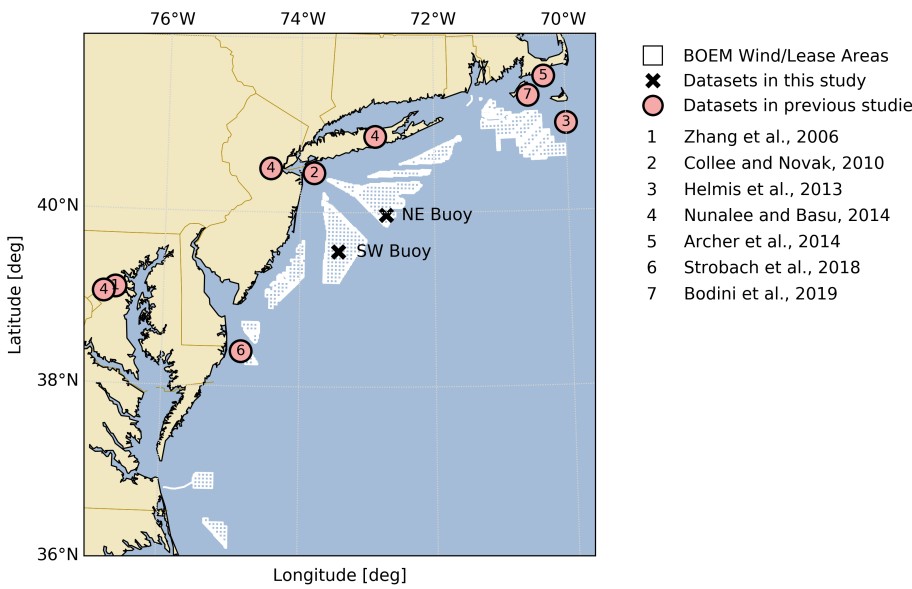

**Figure 1.** Map of U.S. North- and mid-Atlantic OCS showing Bureau of Ocean Energy Management (BOEM) lease areas and wind planning areas in white (accurate as of April 2020), the two floating lidar measurement locations (black crosses) and approximate measurement locations of previous studies focused on the offshore wind resource in this region (red circles).

Extreme wind events relevant to wind turbine operation include rapid changes in flow direction and speed, or persistently high values of shear and veer (Commission, 2019). High vertical wind shear is of particular interest to wind energy as it has a direct effect on wind turbine power and reliability (Colle and Novak, 2010; Peña et al., 2012; Dvorak et al., 2013; Gutierrez et al., 2014, 2017, 2019; Murphy et al., 2019; Borvarán et al., 2020; Hallgren et al., 2020). One phenomenon responsible for producing high-shear events has gained particular attention by the wind energy community: the low-level jet (LLJ), defined as local wind speed maximum in the lower 1000 m of the atmosphere (Soares et al., 2014). Over the last decade, a growing body of work has identified and characterized LLJs within and around current U.S. mid-Atlantic wind energy areas. These offshore LLJs, spanning from Maryland to New Jersey, have been investigated with the Weather Research and Forecasting (WRF) model (Nunalee and Basu, 2014; Colle et al., 2016; Strobach et al., 2018), ship-borne lidar (Pichugina et al., 2017; Strobach et al., 2018), aircraft measurements (Colle et al., 2016), sodar (Helmis et al., 2013), radiosonde (Colle and Novak, 2010; Helmis et al., 2013; Nunalee and Basu, 2014), and radar wind profilers (Zhang et al., 2006; Nunalee and Basu, 2014). A consensus agreement among these studies is the frequent occurrence of persistent LLJs in this area during the warm season.

While some studies were limited to heights above wind turbine operation (Zhang et al., 2006; Nunalee and Basu, 2014), others found wind speed maxima at heights representative of a typical wind turbine rotor (Colle and Novak, 2010; Pichugina et al., 2017; Strobach et al., 2018).

These LLJs are not limited to the U.S. mid-Atlantic, but are a global phenomenon (Parish et al., 1988; Winant et al., 1988; Burk and Thompson, 1996; Parish, 2000; Hoinka and Castro, 2003; Peña et al., 2012; Ranjha et al., 2013; Floors et al., 2013; Peña et al., 2014; Soares et al., 2014; Rijo et al., 2018; Lima et al., 2018; Svensson et al., 2019; Hallgren et al., 2020), occurring both onshore and offshore, and triggered by a range of atmospheric conditions. The most common trigger perhaps is the onset of stable stratification in the lower atmosphere, most commonly at night, which reduces turbulent mixing and allows the expression of the inertial oscillation in the wind profile (Blackadar, 1957; Parish et al., 1988; Parish, 2000; Colle and Novak, 2010; de Wiel et al., 2010). Sloping terrain is also an important driver, where wind speeds closer to the surface accelerate faster than those aloft, producing a LLJ (Holton, 1967; Parish and Oolman, 2010; Du and Rotunno, 2014; Shapiro et al., 2016). Offshore LLJs have been associated with changes in coastal topography (Beardsley et al., 1987; Winant et al., 1988; Strobach et al., 2018), the land-sea temperature gradient (Chao, 1985; Clemente-Colon and Yan, 1999; Colle and Novak, 2010; Floors et al., 2013; Soares et al., 2014).

To date, it is not well-established which of these mechanisms (or combinations thereof) are responsible for LLJs in U.S. mid-Atlantic wind energy areas. This lack of certainty is largely the result of the limited analyses performed to-date. While the aforementioned mid-Atlantic studies (Fig. 1) were extremely valuable in providing an initial characterization of offshore wind conditions, limitations of the measurements used undermine their value to current U.S. East Coast wind energy lease areas. Many of the data sets were spatially disjunct (Colle et al., 2016; Pichugina et al., 2017; Strobach et al., 2018) or limited to coastal areas (Zhang et al., 2006; Helmis et al., 2013; Nunalee and Basu, 2014; Colle et al., 2016). The only two experiments recorded in literature that were far enough from the coast to be representative of conditions that will be experienced by offshore wind plants were limited in duration to a maximum of 1 month (Helmis et al., 2013; Pichugina et al., 2017; Strobach et al., 2018).

Increasing investments in U.S. offshore wind energy along with continuous instrumentation developments have enabled a surge in deployments of offshore wind measurement systems. In particular, the emergence of buoy-mounted floating lidar has led to at least 10 and as many as 20 floating lidar deployments in the U.S. East Coast in recent years. These data have been kept proprietary and any derived analyses have not been disseminated. In August and September 2019, however, the New York State Energy Research and Development Authority (NYSERDA) funded the deployment of two floating lidars (DNV-GL, 2020) within two current lease areas in the New Jersey offshore wind area (Fig. 1). These floating lidars provide wind data at multiple heights across the rotor layer (Table 1). To our knowledge, these deployments provide the first publicly available, and relevant observational data set for the analysis of wind characteristics in U.S. East Coast active lease areas and, as such, are of immense value for wind energy research.

A cursory look at the NYSERDA data alone can reveal very important wind characteristics and phenomena. We show an example of this in Fig. 2, where an intense high-shear event existing over a 2-day period is measured at the northeast (NE) buoy. Not only do we see frequent extreme shear across the nominal rotor area but also several low-level jet (LLJ) events where the

peak in the wind profiles is as low as 100 m. In the highlighted LLJ and monotonic-shear periods, the time-averaged profiles reveal a power-law exponent of 0.59 and 0.32, respectively, when measured across a nominal rotor layer spanning between 40 m and 160 m. This corresponds to wind speed gradient, $\Delta U/\Delta z$, values of 0.12 s$^{-1}$ and 0.08 s$^{-1}$, respectively, across the rotor layer. The ability to accurately predict such events using numerical weather prediction (NWP) models is crucial for wind resource assessment, wind power forecasting, and the timely implementation of operation and maintenance procedures to protect turbines from damage. A proper documentation of these extreme events will help to identify the shortcomings of the models needed for further improvement and will guide the development of more accurate standard guidelines for offshore wind turbines. To our knowledge, the existence of these high-shear events, let alone their causes and development, have not been previously studied in the U.S. East Coast offshore wind lease areas. Our goal is to characterize these events and understand the physical mechanisms governing their onset and dissipation. To do so, we leverage these novel floating lidar observations in the U.S. offshore wind areas.

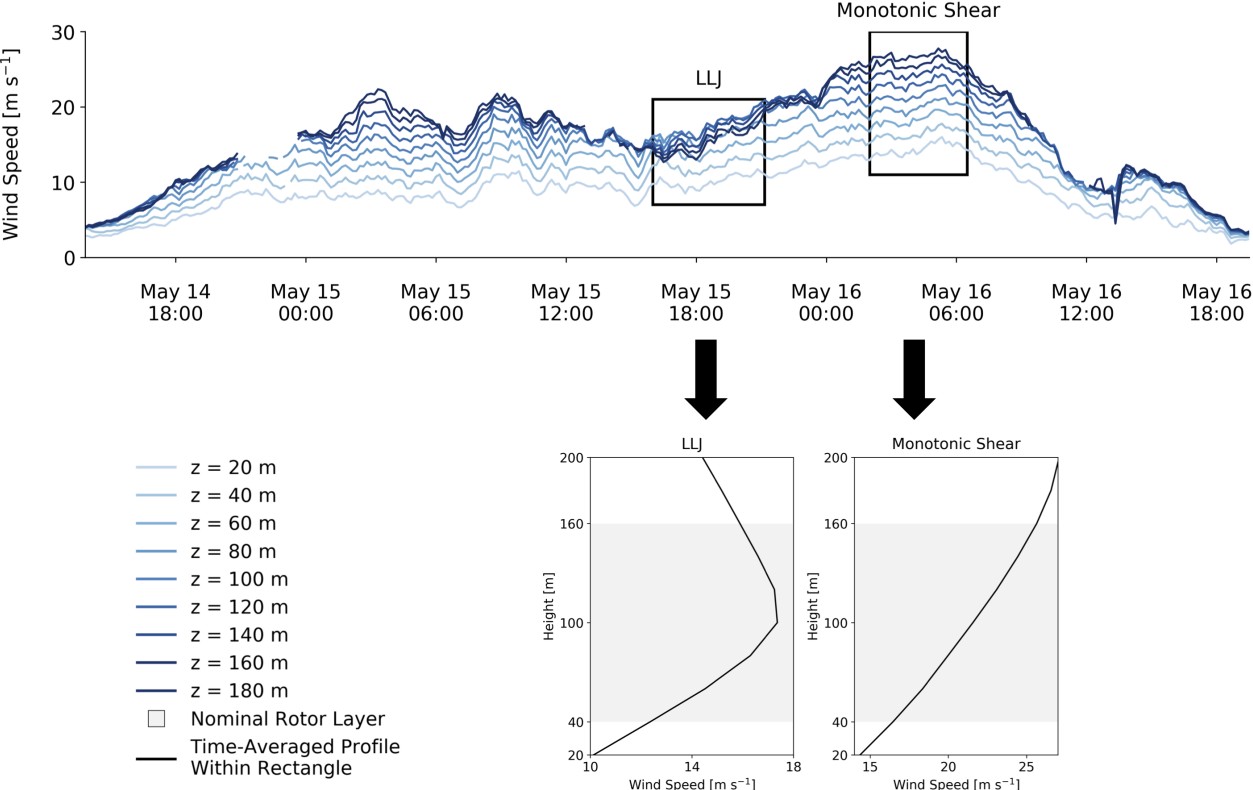

**Figure 2.** Example of high-shear event as measured by lidar on the NE buoy. Location of the NE buoy is provided in Table 1. Time series between May 14, 2020, 13:50 UTC and May 16, 2020, 19:30 UTC (top). The data within the two black boxes are time-averaged and shown below the time series as examples of low-level jet (LLJ) and monotonic-shear periods.

**Table 1.** Summary of data set being analyzed: site name, location (latitude, longitude), period analyzed, distance from coast due west, lidar measurement heights (above mean sea level), and quantities being analyzed.

| Site Name | Location | Period Analyzed | Distance from Coast | Lidar Measurement Heights | Quantities Analyzed |
|---|---|---|---|---|---|
| SW Buoy | 39.55°N, 73.43°W | Sep. 4, 2019 – Aug. 16, 2020 | ~ 69 km | 20–200 m every 20 m | Wind speed and direction, turbulence intensity, 2-m air temperature, sea-surface temperature |
| NE Buoy | 39.97°N, 72.72°W | Aug. 12, 2019 – Aug. 16, 2020 | ~ 114 km | | |

## 2  Identification of High-Shear Events

Time series of vertical profiles of wind speed at the two buoy sites are used to detect and characterize high-shear events that are relevant for offshore wind development. The algorithm developed to detect these events discerns between two types of wind speed profiles: monotonic shear and LLJ (Fig. 2). The algorithm is applied to each 10-minute-mean profile. When high shear is detected for a continuous period of 1 hour or longer, this period is defined as a high-shear event. To avoid double counting, separate events that are close in time and measured at the same site are merged into a single, longer event. This is done in two steps: first, events with lower shear that last 1 hour or less but are sandwiched in between two high-shear periods are identified as an integral part of the adjacent events and merged into them to form one, longer event; finally, two events that are within 6 hours of each other are merged into a single, long-lived event.

The monotonic-shear profiles refer to 10-minute averaged profiles in which the wind speed magnitude strictly increases with height (Fig. 2, right-side profile). For the LLJ cases, the wind speed magnitude increases up to a certain height and then decreases, revealing the presence of a LLJ with a nose below 200 m (Fig. 2, left-side profile). While the monotonic shear cases could be the lower part of a LLJ with a nose above 200 m, the vertical extent of our measurements does not allow for that distinction to be made. For this reason, the algorithm was developed to distinguish between both.

The detection of both types of high-shear profiles is based on several conditions, as outlined below and shown by the schematic in Fig. 3. We define nominal hub height and rotor diameter values to be 100 m and 120 m, respectively (the rotor span being between 40 m and 160 m). These are assumed to be representative of an offshore wind turbine and are used here to facilitate the interpretation of results in the context of offshore wind development. For the analysis performed here, only profiles with a hub-height wind speed greater than 3 m s$^{-1}$ are considered. A profile is classified as "monotonic shear" if

(i) the rotor-layer shear is greater than a prespecified threshold value,

$$\frac{\Delta U}{\Delta z}\bigg|_{\text{rotor}} \geq \frac{\Delta U}{\Delta z}\bigg|_{\text{rotor\_threshold}}.$$

A profile is classified as "LLJ" if

(i) the height of maximum wind speed is between the second (40 m) and second-to-last (180 m) measurement height,

$$40\ m \leq z\,(U_{\mathrm{max}}) \leq 180\ m;$$

(ii) the wind speed gradient between the rotor bottom and the nose height ( $\left.\frac{\Delta U}{\Delta z}\right|_{\mathrm{nose}}$ ) is greater than the same pre-specified threshold value used for the monotonic-shear detection,

$$\left.\frac{\Delta U}{\Delta z}\right|_{\mathrm{nose}} \geq \left.\frac{\Delta U}{\Delta z}\right|_{\mathrm{rotor\_threshold}} ;\ \text{and}$$

(iii) the wind speed drop off above the jet nose meets minimum requirements in terms of dimensional and dimensionless threshold values,

$$\Delta U_{\mathrm{drop}} \geq 1.5\ \text{m s}^{-1}\ \text{and}\ \frac{\Delta U_{\mathrm{drop}}}{U_{\mathrm{nose}}} \geq 10\%$$

where $\Delta U_{\mathrm{drop}} = U_{\mathrm{top}} - U_{\mathrm{nose}}$ and $U_{\mathrm{top}}$ marks the top of the jet and is the first local minimum in wind speed identified above the nose. If a minimum is not found, a jet nose cannot be identified and the profile is not flagged as a LLJ. Depending on the threshold of the wind speed drop, $\Delta U_{\mathrm{drop}}$, the number of the detected events can vary (Kalverla et al., 2019). For most of the analysis in Kalverla et al. (2019), the threshold used for $\Delta U_{\mathrm{drop}}$ is 2 m s$^{-1}$. The enforcement of both dimensional and nondimensional wind speed drop off criteria is based on previous work (Baas et al., 2009) but the threshold values are adjusted in magnitude here because of the limited vertical extent of the measurement data available.

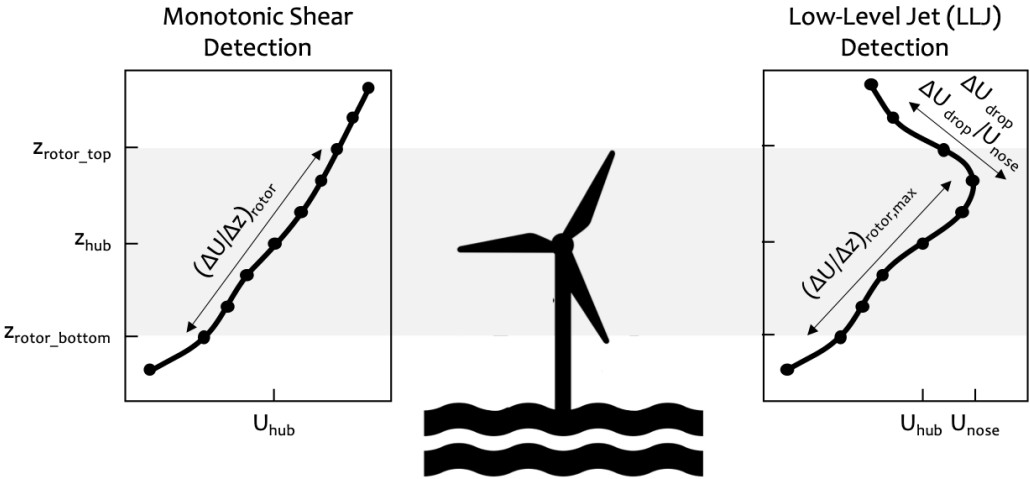

**Figure 3.** Schematic showing key quantities used in the algorithm developed to detect the two types of high-shear profiles considered herein: monotonic shear and low-level jet (LLJ). Individual detections are then merged into events.

In the wind energy industry, the vertical wind shear is typically represented by the power-law exponent, $\alpha$ (Commission, 2019). However, in this work, the variable used to quantify vertical wind shear is wind speed gradient between a reference

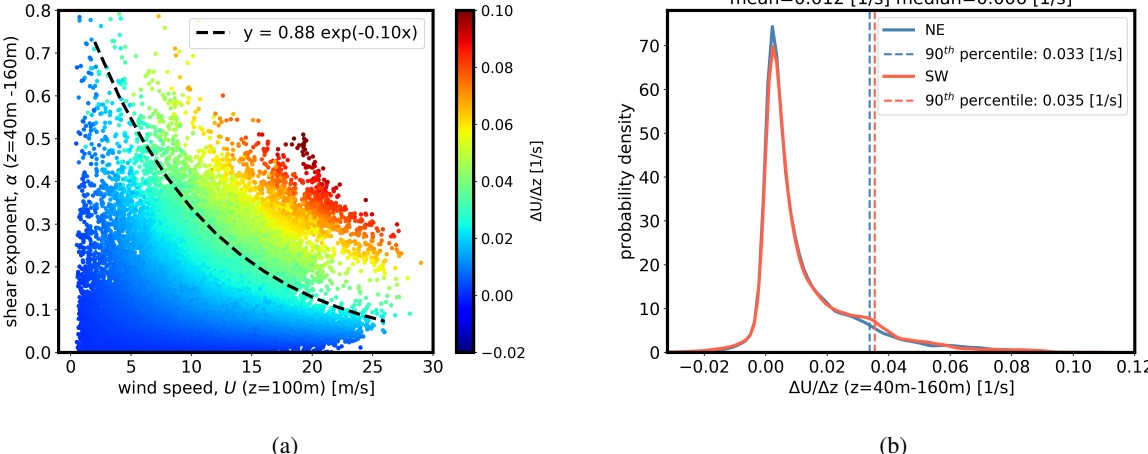

(a)            (b)

**Figure 4.** An analysis of vertical wind shear parameters from the NE floating lidar. Panel (a) shows the relationship between the wind speed at 100-m height ($U_{100m}$), the power-law exponent between height 40 m and 160 m ($\alpha$), and the wind speed gradient between 40 m and 160 m ($\Delta U/\Delta z$). The black dashed line represents the $90^{th}$ percentile value of $\Delta U/\Delta z$. Panel (b) shows the probability distribution of $\Delta U/\Delta z$ for both buoys.

height (here taken as 40 m) and other heights above it. A relationship plot (Fig. 4a) among wind speed at hub height, $U_{100m}$, wind speed gradient across the rotor, $\frac{\Delta U}{\Delta z}$, and shear exponent, $\alpha$, explains that the shear exponent can be very low even though a turbine faces a high wind speed difference across its diameter. The shear exponent is nondimensional and does not consider the magnitude of wind speed that a turbine actually faces. As a result, data points that would normally be considered as high shear by $\alpha$ often have relatively low wind speeds and would not pose a danger to wind turbines. The fitted black dash 130   line (Fig. 4a) provides the change of extreme wind shear exponent with wind speeds rather than a constant shear exponent threshold (e.g., 0.2). It explains that the threshold for the extreme wind shear exponent should decrease with an increase of wind speed to properly consider the wind speed gradient across the rotor diameter. To better capture events that do pose that danger, we consider instead the $\frac{\Delta U}{\Delta z}$ metric—which does account for wind speed magnitude—as a threshold for detecting high wind shear events. The distribution of $\frac{\Delta U}{\Delta z}$ for the buoys are presented in Fig. 4b. The figure shows a long tail in the distribution 135   that captures a considerable number of high shear events. Setting a threshold at the $90^{th}$ percentile, as shown in the figure, captures a large number of events while ensuring that the shear values are extreme. Herein, for both types of profiles, the threshold shear value, $\frac{\Delta U}{\Delta z}\big|_{\text{rotor\_threshold}}$, is set to the $90^{th}$ percentile of the distribution of $\frac{\Delta U}{\Delta z}\big|_{\text{rotor}}$ over the entire measurement period, which equals 0.035 s$^{-1}$ (Fig. 4b) when averaged across the lidars. Note that we are using fixed heights (e.g., 40 m to 160 m) to calculate the wind shear exponent and wind speed gradient across the rotor. However, the wind shear exponent and 140   wind speed gradient will be underrepresented across the rotor for the LLJ cases which have wind speed maxima below 160 m height.

# 3 Results

## 3.1 Detected Events

We first summarize the results of the high-shear detection algorithm in Fig. 5. A large number of events are detected at both lidars, most of which are less than 10 hours but some that extend for more than two days. All the events identified based on the detection criteria are marked as "high shear" events. The events presented in this section include both LLJ and monotonic-shear cases. The total number of detected events are 104 and 92 for the northeast (NE) and southwest (SW) buoy, respectively. To explain why there are more events at the SW buoy, we must first better understand the atmospheric conditions in which these events are able to occur. We begin this investigation in the next section by looking at seasonal and diurnal trends in event frequency.

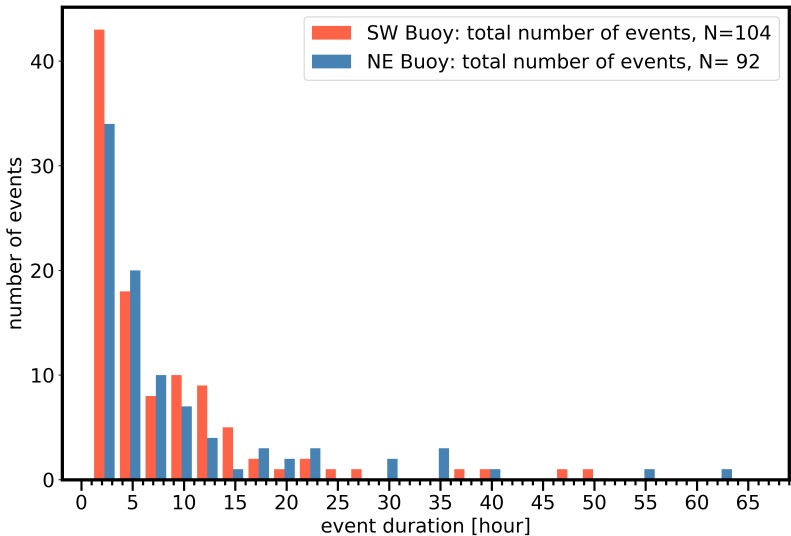

**Figure 5.** Number of high shear events for both buoys as a function of event duration. Only events with a minimum duration of 1 hour are considered.

## 3.2 Seasonal and Diurnal Dependence

We explore seasonal and diurnal trends in the high-shear events in Fig. 6. In Fig. 6a and Fig. 6b, we consider the number of 10-minute average data points as they depend on hours of diurnal cycles and months, respectively. In Fig. 6c, we consider actual event counts by month. Fig. 6a shows a clear diurnal trend in the high-shear events, with event frequency increasing after noon and dropping after 22:00. Indeed, events are twice as likely to happen during the night than during the morning. We see in Fig. 6b and Fig. 6c that there is also a strong seasonal trend in event frequency. Events are largely concentrated in the

spring months (i.e., March through June) and are much less frequent in the rest of the year. In particular, the month of June has the highest number of events (16 events, on average) and November has the lowest number of events (one event, on average).

The presence of strong diurnal and seasonal trends in the number of high-shear events suggest the influence of meteorological conditions. Indeed, we expect this to be the case that follows the well-established relationships between high wind shear, LLJs, and thermodynamic atmospheric stability established by previous works (Sergeevich and Obukhov, 1954; Blackadar, 1957; Holton, 1967; Stull, 1988; Burk and Thompson, 1996; Parish, 2000; Poulos et al., 2002; Wharton and Lundquist, 2012; Ranjha et al., 2013). In the next section, we explore this possible relationship between high shear and atmospheric stability in more detail.

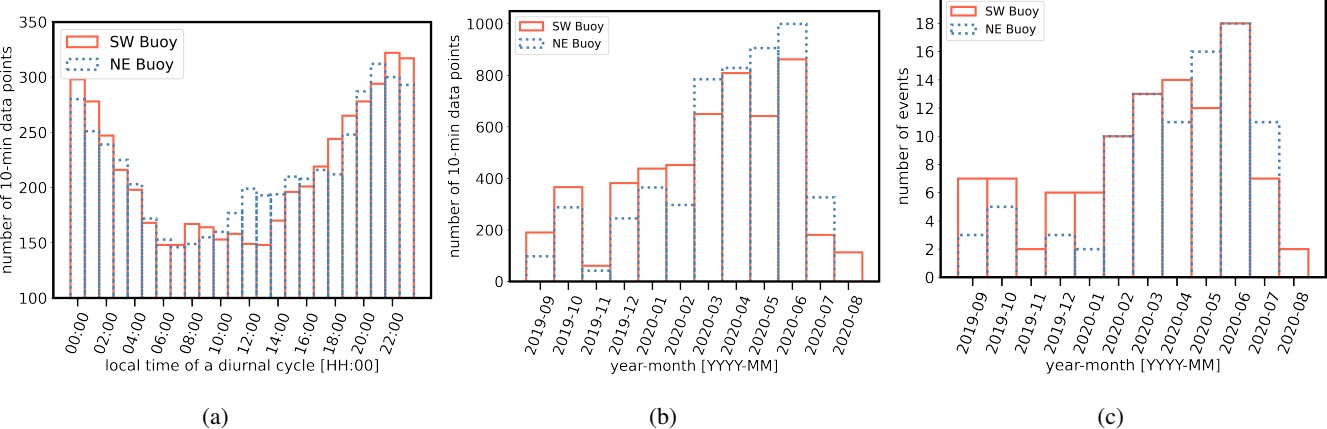

**Figure 6.** Diurnal and seasonal distribution of high-shear events at both buoys: number of 10-min profiles in which high shear was detected as a function of local time (a) and month (b), and number of events across the year (c).

## 3.3 Atmospheric Stability and Turbulence

In this section, we intend to investigate the relationship among the high-shear events, atmospheric stability, and turbulence. However, we do not have air temperature measurements at different heights to appropriately characterize the atmospheric stability. Instead, we use the difference between 2-m air temperature and the sea-surface temperature as our best proxy for atmospheric stability. We herein denote this air-sea temperature difference as $\Delta T$. Of course, the air-sea temperature difference is more of an external forcing to the atmosphere, but it may provide some indication of atmospheric stability, such as when warm air flows over a colder sea inducing a stable stratification. To measure turbulence, we use the turbulence intensity (TI) measurements at 100 m as measured by the floating lidars, denoted as $TI_{100m}$. Turbulence intensity is defined as standard deviation of the wind speed normalized by the mean wind speed of the 10-minute window. In Fig. 7a & Fig. 7b, we plot distributions of $\Delta T$ and $TI_{100m}$, where the full data set are shown in blue with the high-shear events shown in orange.

It is clear from Fig. 7a that high-shear events are strongly associated with a positive air-sea temperature difference ($\Delta T > 0$). The distribution of $TI_{100m}$ is shown for both high-shear events and the full data set in Fig. 7b. The high shear events have

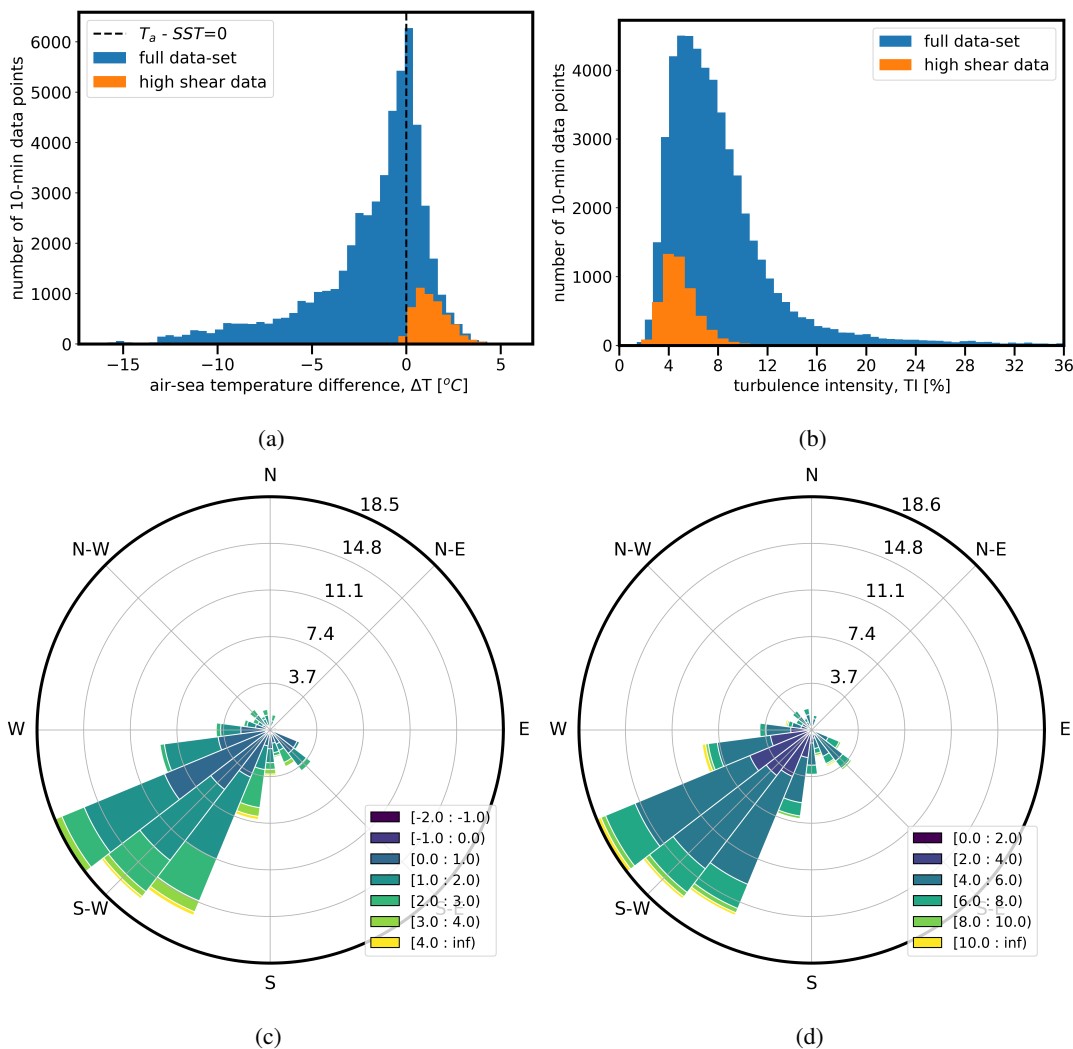

**Figure 7.** The impact of air-sea temperature difference on wind shear and turbulence intensity. Top figures (a, b) show distribution for entire data set vs. periods in which high shear was detected. Bottom figures (b, d) show distribution roses for high-shear periods only. Data are shown for NE buoy only.

turbulence intensity mostly within the bin of 4% to 6% (mean $TI_{100m}$, 5.1%), whereas mean turbulence intensity of all the data sets is 8.3%. Focusing only on the high-shear events (i.e., the orange distributions), we plot $\Delta T$ and $TI$ distributions by wind direction in Fig. 7c and Fig. 7d. We see that these high-shear events are almost exclusively associated with southwesterly flow with a mean wind direction of 217$^o$. Referring to Fig. 1, we see that southwesterly flow is about parallel to the coastline and features an area of very large ocean fetch. The coastline parallel flow has also been identified in previous works (Winant et al., 1988; Hoinka and Castro, 2003; Colle and Novak, 2010; Soares et al., 2014). Although we can't provide an explanation of this coastline parallel flow due to the limitations of the measurements used in this study, these previous studies have explained this

particular flow direction based on detailed observational and numerical model data. The coastal flows are influenced by the high

pressure system over the ocean and a low pressure system inland induced by a sharp contrast between high temperature over land and lower temperature over the sea (Winant et al., 1988; Hoinka and Castro, 2003; Soares et al., 2014). The coast-parallel flow is then generated by the geostrophic adjustment and deflection due to the Coriolis force (Soares et al., 2014).

The observations in Section 3.3 suggest a positive correlation between the near surface temperature gradient and these high-shear events. Depending on the locations, there are several factors such as topography (Winant et al., 1988), thermal

forcing over sloping terrain (Holton, 1967) can facilitate the LLJ occurrence. Blackadar (1957) explained that LLJs are inertial oscillations in the wind triggered by the rapid reduction in surface stress (e.g., frictional decoupling) in the boundary layer. It is possible that warmer air coming from the southwest encounters the colder waters off the mid-Atlantic causing a positive air-sea temperature difference. This temperature difference would then induce stable stratification where vertical turbulent exchange from surface winds to those aloft would be reduced and a degree of "decoupling" of winds aloft from the surface would occur.

Combined with the long ocean fetch where surface roughness is low, this is likely leading to very low turbulence in the winds aloft at the floating lidars, sufficient to cause high wind shear and allow for the formation of low-level jets.

We provide evidence of this induced stratification in Fig. 8 for two high-shear events. As shown for both case studies, the onset of high shear aligns with the switch from a negative $\Delta T$ to a positive $\Delta T$ value. Notably, the end of the second high-shear event (e.g., event-02) aligns with the switch back to a negative $\Delta T$ value and a sharp change of wind direction. The

sharp change in air-sea temperature difference and wind direction suggest the evidence of a frontal passage within this event. The wind direction change in the "event01" is not as sharp as the "event-02" but well-correlated with the change of air-sea temperature difference. Furthermore, we see that the change in sign in $\Delta T$ is driven by changes in the air temperature, $T_a$, whereas the $SST$ remains relatively constant before, during, and after the high-shear events. So indeed, the arrival of warm air from the southwest and the resulting induction of stable stratification appears to be a dominant contributor to these high-shear

events.

We further examine the role of the air-sea temperature difference in influencing wind conditions in Fig. 9. Here, we consider the full set of data and not just the high-shear events. Specifically, we show the relationship between $\Delta T$ and wind speed at 100 m, TI at 100 m, the shear exponent, $\alpha$, across the rotor layer, the maximum wind speed gradient across the rotor, $\Delta U/\Delta z_{max}$, and wind veer. The data are bin-averaged and shown along with the standard deviation within the bin. The

density of the data is shown in red in the background. We observe that wind speed at hub height is almost constant when the temperature difference is negative, but increases sharply when the temperature difference is positive. The linear increase of wind speed with an increase of positive temperature difference ($\Delta T > 0$) suggests that the strength of extreme events is highly dependent on the magnitude of positive temperature difference. On the other hand, turbulence intensity at hub height drops as the temperature difference approaches zero, showing a strong dependency on static stability (Fig. 9b). There is an upward trend

in the turbulence intensity after $\Delta T = 2\ ^oC$. This could be caused by a low density of the data within the bin. Similar to wind speed, the shear exponent (Fig. 9c), the maximum wind shear (Fig. 9d), and wind veer (Fig. 9e) are roughly constant when $\Delta T$ is negative before increasing sharply when the difference becomes positive. As both wind shear and veer increase with positive $\Delta T$, any possible relationship between the wind shear and wind veer is investigated in Fig. 9f. It is observed that the wind veer

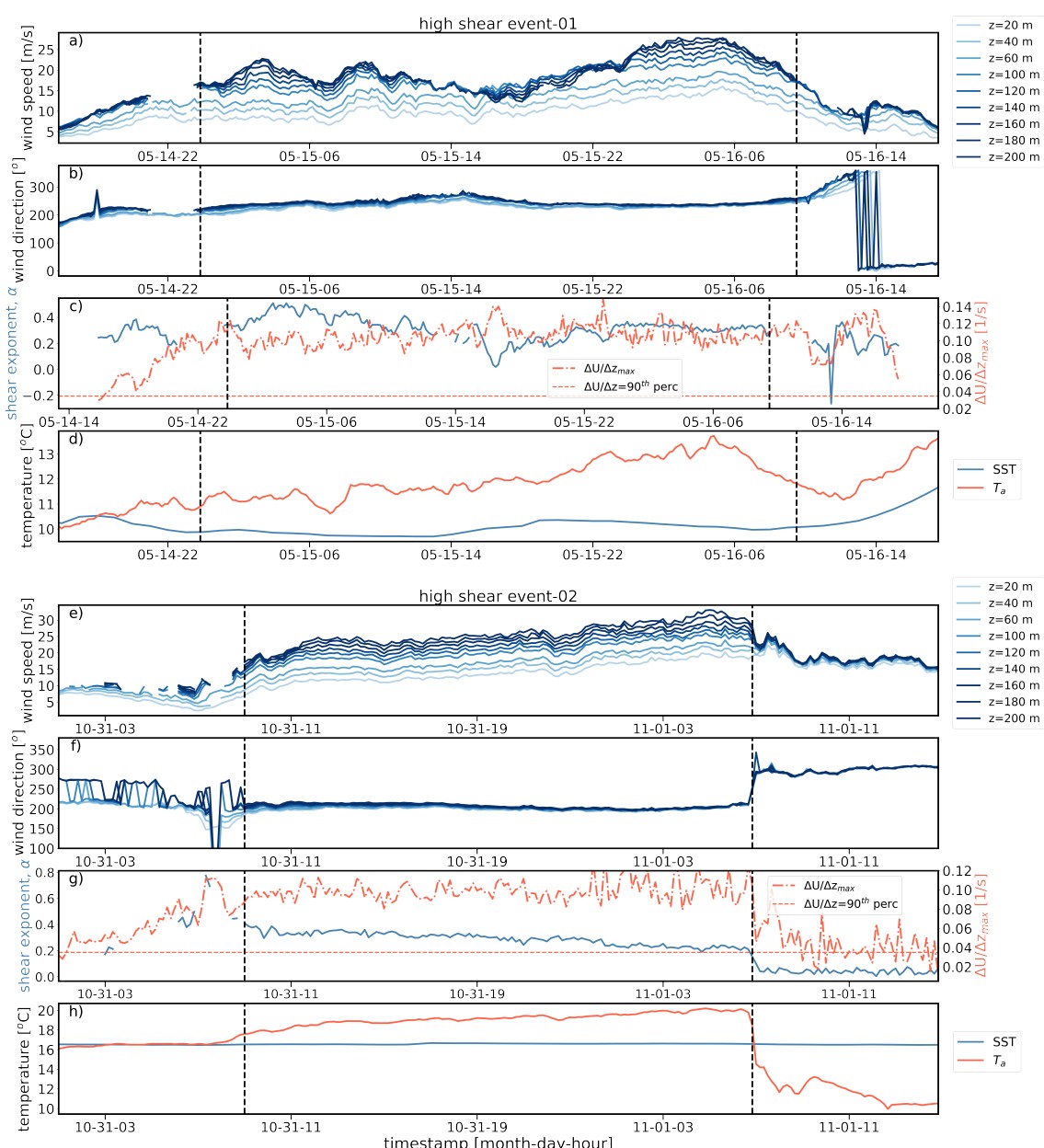

**Figure 8.** Two examples of high-shear events measured at the NE buoy (Table 1) in year 2019. Subfigures show time series of wind speed (a, e), wind direction (b, f), wind shear (power-law exponent $\alpha$), and wind speed gradient ($\Delta U/\Delta z_{max}$) (c, g), and air ($T_a$) and sea-surface temperature (SST) (d, h). Vertical dashed lines represent the start and end times of the high shear events. First example is shown with subfigures a, b, c, d, and second example is shown in subfigures e, f, g, h.

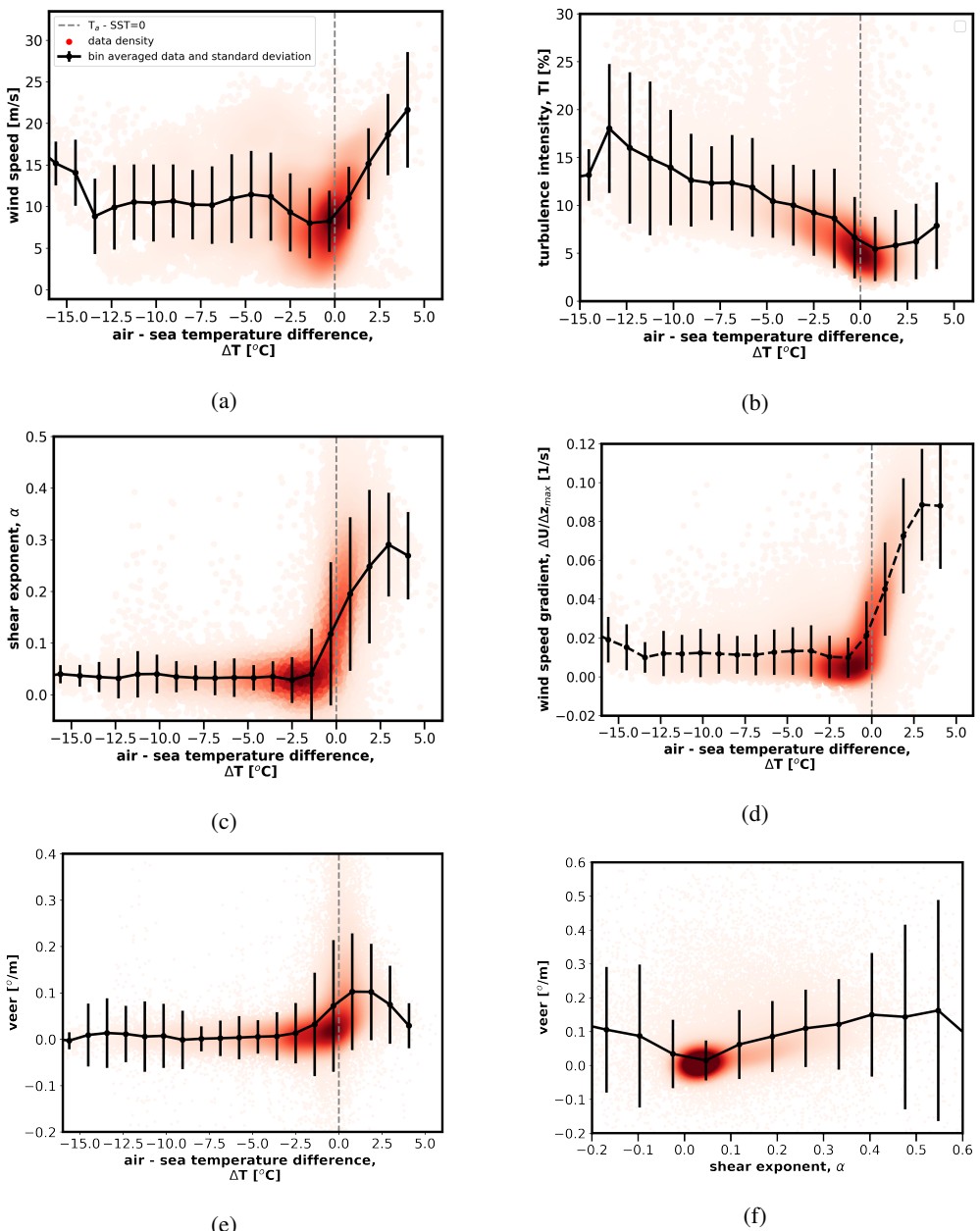

**Figure 9.** Wind characteristics as they depend on the air-sea temperature difference. Only NE buoy (Table 1) data are shown here. a) Wind speed, b) turbulence intensity, c) wind shear exponent, d) maximum wind speed gradient across the nominal rotor defined here (between 40 m and 160 m), and e) wind veer. A relationship between the wind shear exponent and wind veer is provided in subfigure f.

increases with an increase of wind shear. The upward trend of the wind veer when the wind shear exponent is negative caused 220 by a low density of the data. Similarly, we are not confident in the relationship above wind shear exponent 0.4. Note that, as

the wind direction is calculated with the ratio of wind speed components, wind shear exponent is better suited than wind speed gradient to show the relationship between the wind veer and wind shear.

## 3.4 Spatial Variability

In this section, we briefly explore the potential reasons for having 13% additional events at the SW buoy over the NE buoy. The two buoys are located at different locations in the wind lease areas and vary in proximity to the coast (Table 1). We can apply these findings to inform developers on the different conditions that can be expected in these regions within the lease areas. The high shear events occur with the southwesterly flow described in Section 3.3. Thus, wind farms installed in the SW region of the lease areas will be impacted by the southwesterly winds before the wind farms installed close to the NE buoy.

In Table 2, we compare mean atmospheric variables between the two buoys, both for the high-shear cases and the full data set. To perform a proper intercomparison between the buoys, we only considered time stamps that are common for both buoys. Table 2 shows that the local air temperature at the NE buoy is lower than the SW buoy. Furthermore, the difference of air temperature between the buoys, $T_{a,SW}$-$T_{a,NE}$, is higher than the change of SST between the buoys, $SST_{SW}$ -$SST_{NE}$. Therefore, the lower air temperature at the NE buoy is largely responsible for its lower air-sea temperature difference relative to the SW buoy. This higher air-sea temperature difference at the SW buoy corresponds to notably lower TI and a slightly higher wind speed gradient across the rotor relative to the NE buoy. The SW and NW buoys are $\sim$ 69 km and $\sim$114 km far from the coast, respectively. The SW buoy which is closer to the coast faces higher air-sea temperature difference than the NW buoy. It suggests that the coast has an impact on the buoys and the impact varies depending on the distance from the coast.

**Table 2.** Comparison of the mean atmospheric variables between the NE and SW buoys.

| Variables [units] | High shear data | | | All the data | | |
|---|---|---|---|---|---|---|
| | SW buoy | NE buoy | SW buoy - NE buoy | SW buoy | NE buoy | SW buoy - NE buoy |
| $T_a$ [$^oC$] | 13.97 | 13.62 | 0.35 | 13.01 | 12.67 | 0.346 |
| $SST$ [$^oC$] | 12.45 | 12.21 | 0.23 | 14.58 | 14.60 | -0.023 |
| $T_a$-$SST$ [$^oC$] | 1.52 | 1.4 | 0.12 | -1.62 | -1.94 | 0.31 |
| $\alpha$ [ ] | 0.286 | 0.289 | -0.003 | 0.103 | 0.097 | 0.0066 |
| $\Delta U/\Delta z$ [s$^{-1}$] | 0.050 | 0.0491 | 0.0010 | 0.0127 | 0.0123 | 0.0004 |
| $\Delta U/\Delta z_{max}$ [s$^{-1}$] | 0.0753 | 0.0731 | 0.0022 | 0.0250 | 0.0245 | 0.0005 |
| $U_{100m}$ [$m\ s^{-1}$] | 16.179 | 15.735 | 0.445 | 9.843 | 10.116 | -0.2736 |
| $TI_{100m}$ [%] | 4.379 | 5.119 | -0.740 | 7.833 | 8.327 | -0.4930 |

## 3.5 Low-Level Jets

Up to this point, the analysis considered high-shear events irrespective of the profile characteristics across a nominal rotor span. Here, we focus on a subset of 10-minute periods that are interspersed within these high-shear events: those with a LLJ. These

events are of particular interest to wind energy applications as they subject the rotor not only to high shear, but also to negative shear when the jet nose is within the rotor span.

Out of the 104 (92) high-shear events detected for the SW (NE) buoy, 30% (26%) feature LLJs and 9% (7%) are made up entirely of LLJ profiles. These profiles were not detected at any specific point of the high-shear events. Instead, they occurred at the beginning, end, and throughout the longer-lived events. A simple statistical analysis of these LLJ profiles confirms that they are highly relevant for wind turbine operation: the most common nose wind speeds are between 9 m s$^{-1}$ and 12 m s$^{-1}$, and the most common nose heights are 80 m and 100 m. As expected, the predominant wind direction during these LLJ occurrences is consistent with that for the long-lived, high-shear events: primarily from the SW sector. These LLJs exhibit a clear seasonal signature, being most frequent in spring and not occurring at all in winter (Fig. 11a). No clear diurnal signature for these LLJ events can be identified from Fig. 11b. It should be noted that this study uses a year of observational data, but multi-year of data would be more useful to investigate the seasonal variability and climatology.

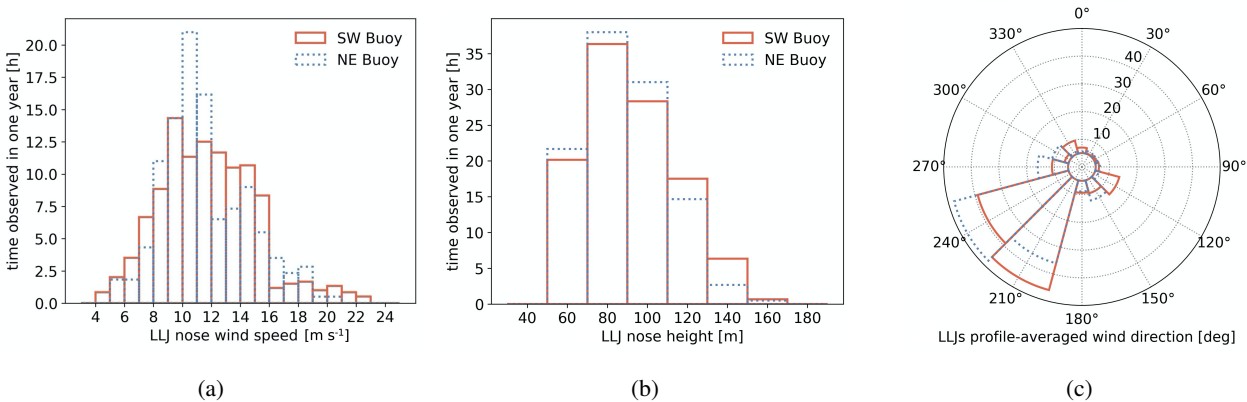

**Figure 10.** Number of hours of LLJ as a function of nose wind speed (a), nose height (b), and vertically averaged wind direction (c). Distributions consider all 10-minute profiles featuring a LLJ.

The highest shear values observed throughout this year of measurements correspond to LLJ profiles, as evidenced by the pronounced tail of the LLJ maximum-shear distributions in Fig. 12a. When the nose of the jet is within the rotor swept area, a portion of the rotor will experience negative shear. Here, we quantify how much of the rotor experiences negative vs. positive shear for each LLJ profile using the turbine-jet relative distance parameter [$\xi$, Gutierrez et al. (2017, 2019)]. These values are shown in Fig. 12b: -1 indicates entirely positive shear across the rotor, 0 half negative and half positive, and 1 entirely negative. This analysis reveals that the nominal rotor defined here experiences at least some negative shear during most of the LLJ profiles identified: less than 1% of LLJs have $\xi = -1$. More than 50% of the LLJ profiles identified have more negative than positive shear across the rotor ($1 > \xi > 0$). While the mean negative shear is not too high (i.e., $\Delta U/\Delta z$=-0.024 s$^{-1}$ for both buoys), the distribution reveals a noticeable tail where $\Delta U/\Delta z < $ -0.035 s$^{-1}$ (Fig. 12c). While previous work (Gutierrez et al., 2017) has found that negative shear can decrease loads on the wind turbine system (primarily at the nacelle and tower),

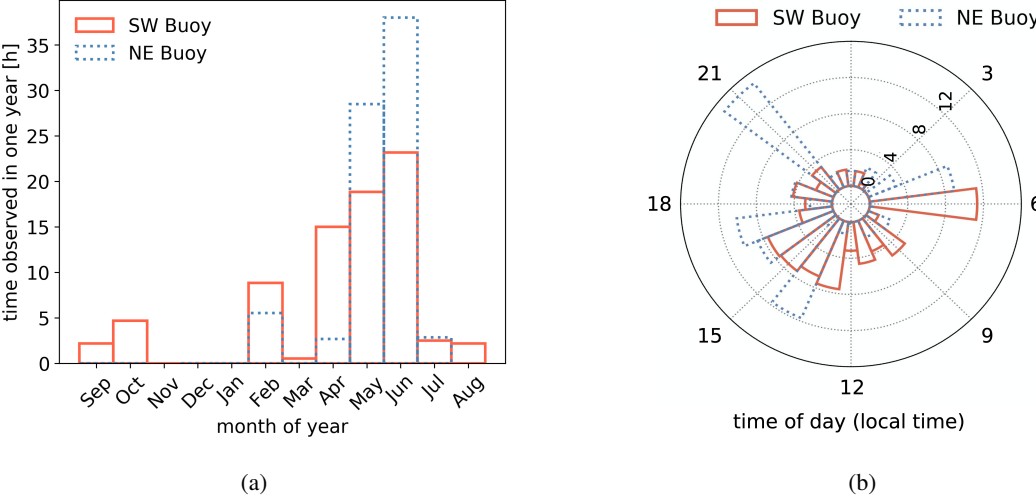

(a)                  (b)

**Figure 11.** Number of hours (over the entire year) in which LLJs were observed at both buoys as a function of month (a) and time of day (b). Distributions consider all 10-minute profiles featuring a LLJ.

the positive shear in these profiles has been directly linked to an increase in static and dynamic loads relative to a well-mixed profile (Gutierrez et al., 2016). A recent study (Gutierrez et al., 2019) investigated the symmetry in wind turbine loads when the rotor experiences half-positive, half-negative shear and found complex interplay between the tower, blades, and gravitational

loads. The complexity of this aero-structural problem and the nature of these boundary layer profiles off the U.S. East Coast highlight that more studies are needed to support the successful deployment of offshore wind turbines in the United States.

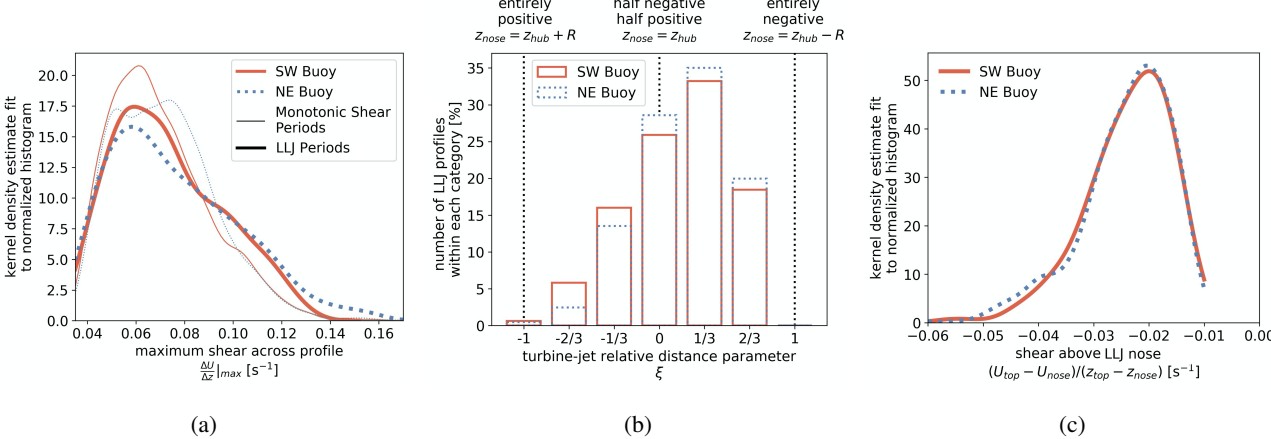

(a)                 (b)                 (c)

**Figure 12.** Distribution of maximum shear over 1 year of measured profiles and separated by profile type: monotonic vs. LLJ (a); distribution of turbine-jet relative distance parameter for all LLJ profiles (b); distribution of shear above LLJ nose (between nose and local wind speed minimum measured above it), shown only for 10-minute periods with LLJ profile (c).

The high-shear periods measured at the two sites had substantially lower turbulence levels than the remainder of the data. This is exemplified in Fig. 13, wherein TI is given as a function of wind speed for all 10-minute periods without a high-shear profile (black) and those with a LLJ profile (colors) only. Note that the monotonic shear profiles are not included here, but their turbulence distribution is similar to that of the LLJ profiles. As expected, most of the data (the profiles not flagged as having high shear) follow a decreasing trend with wind speed up to a certain point, and then see a slight increase as wind speeds go up again and generate mechanical turbulence. For example, the SW buoy goes from 5.9% TI at 8 m s$^{-1}$ to 7.8% TI at 20 m s$^{-1}$. The same is not observed for the LLJ-exclusive data: a TI value of 4.9% at 8 m s$^{-1}$ decreases even further as the wind speed increases, to about 3.7% at 20 m s$^{-1}$. This is likely connected to stable atmospheric stratification, which has been found to support LLJ formation and suppress turbulence not only on land but offshore in the U.S. East Coast (Colle and Novak, 2010).

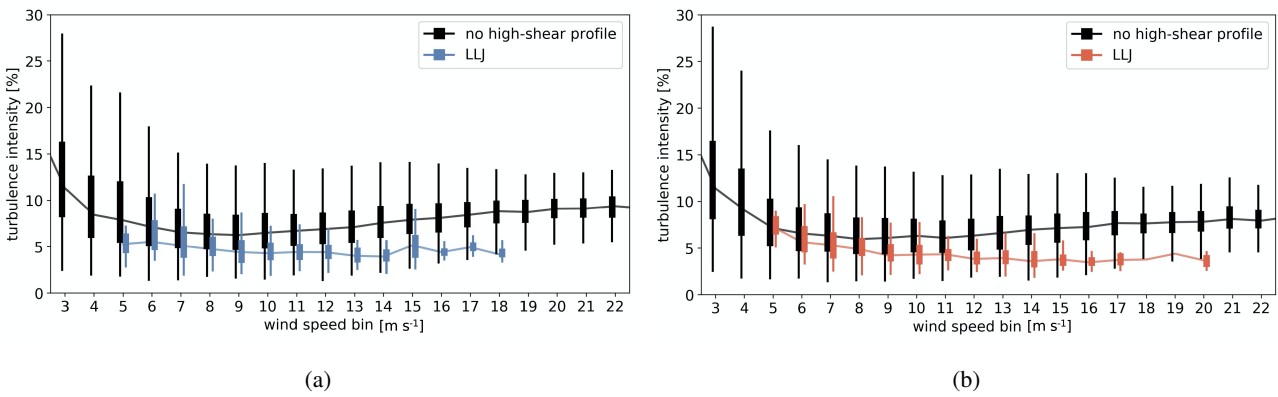

(a)                                                          (b)

**Figure 13.** Hub-height ($z = 100$ m) distribution of turbulence intensity for wind speed bins between 3 and 22 m s$^{-1}$ for the NE (a) and SW (b) buoys. Distributions are shown separately for all 10-min periods without a high-shear profile (black) and those with a LLJ profile (colored). Only wind speed bins with at least 10 LLJ profiles are shown. Monotonic shear periods are excluded here for clarity.

## 4    Synoptic Overview

Our analysis to this point has demonstrated the frequency of extreme high-shear events that are associated with stable stratification induced by warmer air from the southwest flowing over colder mid-Atlantic waters. In this section, we examine synoptic charts from NOAA's Weather Prediction Center archive (https://www.wpc.ncep.noaa.gov) for each case to examine the synoptic conditions that lead to the arrival of warmer southwest air.

Synoptic conditions during these high-shear events generally include a surface low-pressure system centered west of the floating lidar locations and a region of high-pressure to the east, as depicted in the schematic shown in Fig. 14a. This schematic is a generalization of the synoptic setup for roughly 75% of the 86 days that registered an event. The exact location and strength of these pressure systems deviates from case to case but the general pattern holds, resulting in a large southerly component to the near surface winds. Due to the differences in location and strength of these pressure systems, a composite schematic was

avoided as the averaging would generate a diffuse depiction of the environment. The directional component of the wind speeds is an important feature as winds coming from the south typically result in warmer air being advected into the area. Additionally, winds with a southwesterly component may be coming from onshore and can contain much higher air temperatures because of stronger heating over land during the day. Further, the long fetch over the ocean results in low turbulence conditions.

This synoptic setup has been observed in previous studies pertaining to offshore low-level jets in the mid-Atlantic region, such as Zhang et al. (2006), Colle and Novak (2010), Helmis et al. (2013), and Strobach et al. (2018). While these studies each provide different mechanisms for the low-level jet formation (such as downslope winds from near-shore topography, differential heating over land and sea, sloping marine boundary layers, cold water upwelling, etc.), the synoptic setups from each study are generally consistent with each other. In most cases, the cyclone to the west advances toward the east or northeast

denoted by the blue arrow in Fig. 14a.

Many of the stronger events coincide with the western low-pressure system strengthening and moving eastward as the pressure gradient ahead of the cold front tightens and increases the wind speeds over the floating lidars (see Fig. 14b). Of the 10 longest events (averaging 30 hours in duration), 7 exhibited a tightening of the gradient and increase in wind speed as the event progressed. Helmis et al. (2013) and Strobach et al. (2018) found a similar tightening of the pressure gradient

during cases of offshore low-level jets in the mid-Atlantic resulting in a strengthening of the wind speeds and shifting of the winds to contain a stronger westerly component. Interestingly, the western low-pressure systems in the two longest events were associated with named winter storms (Isaiah and Ruth, respectively). In fact, 12 out of 16 named winter storms that impacted the East Coast were also associated with high-shear events giving credence to the idea that strong low-pressure systems over the contiguous United States may produce the synoptic setup required for these offshore high-shear events. Expanding to consider

the 25 longest events (averaging 19 hours in duration) shows that only 12 exhibit this eastward propagation and deepening of the low-pressure system. This implies that while the advancing and strengthening low-pressure system is common in the longest events in this area, it may not be a good characterization of all events including those with a much shorter duration.

Lastly, many of the events end around the time of frontal passages as depicted in Fig. 14c. This can be seen in Fig. 8 (event-02) where a sharp drop in temperature (bottom panel, h) coincides with a drastic decrease in shear across the rotor plane (middle

panel, g). The wind shift from south-southwesterly to west-northwesterly is also shown (Fig. 8f) as would be expected during a typical cold frontal passage at this location. This results in colder, well-mixed air advecting over the relatively warmer sea surface temperatures and breaks up the stable conditions favorable for generating high-shear. On the other hand, the majority of events – such as the event shown in Fig. 8 (event-02, e-g)—end well after frontal passage or have no clear synoptic event that can be attributed to the demise of the high-shear. Of the 25 longest events, seven show the ends attributed to frontal passage

(one warm front, 6 cold fronts); however, five of these events are within the 10 longest duration events. While this is clearly not applicable to the majority of events, many events, especially those that are around 6 hours or less in duration, are difficult to determine how the event ends as the synoptic charts are output at 6 hour intervals. Other noticeable features that were seen in the synoptic charts around the time an event ended were stationary fronts or shortwave troughs (which are relatively small scale synoptic disturbances commonly associated with changes in wind direction near the surface but no, or slight, changes

in temperature). Additionally, some events are considered to have "begun" or "ended" erroneously due to missing data either

before or after the event, respectively. In these cases, it is not possible to determine the physical process that produced or destroyed the high-shear event.

There are no clear synoptic differences between the LLJ events and monotonic shear events. This may be due to the limited observational height at which jet noses above 180 m cannot be determined. It is possible that some events that are considered high shear are, in fact, LLJs with noses above 180 m. Additionally, it is possible that only subtle differences in the air temperature, wind speed, and/or wind direction are able to augment the wind profile such that an LLJ nose develops, or doesn't develop, below 180 m.

For the event days that did not display the setup illustrated in Fig. 14 (roughly one quarter of event days), 13% displayed synoptic conditions with a surface high-pressure system over the coastal mid-Atlantic region and offshore lease area. This results in weak synoptic flow over the offshore lease area and conditions greatly subject to diurnal processes. A similar synoptic environment is found in a case study within Nunalee and Basu (2014) where daily low-level jets formed in coastal New Jersey under an area of high-pressure centered just offshore of the mid-Atlantic states. Additionally, one event occurred as Tropical Storm Arthur approached the lidars from the south off the coast of South Carolina and moved north-northeast. Wind directions, in this case, were from almost directly east, however, air temperatures became warmer than the sea surface temperature as the high-shear event began. From this, it becomes apparent that warm air advection over relatively colder water is an essential ingredient to the formation of these high-shear events that is typically caused by flow with a large southerly component.

## 5  Conclusions

This study has revealed the frequent occurrence of extreme high shear events in U.S. mid-Atlantic offshore wind lease areas. These events were characterized based on data from two floating lidars recently deployed by NYSERDA. We identified approximately 100 high-shear events over a year, with some events lasting up to 3 days. The magnitude of these events was striking, with maximum and mean hub-height wind speeds of 33 m s$^{-1}$ and 16 m s$^{-1}$, respectively, and maximum and mean of power-law wind shear exponent across the rotor of 0.82 and 0.28, respectively. These values are substantially higher than 0.2, the number proposed in the design standards to identify extreme shear conditions relevant to turbine operation (Commission, 2019). It is clear that once wind farms are built in these areas, these extreme events will have substantial effects on wind turbine power generation and structural response.

Fortunately, these extreme events seem to be fairly predictable. We found that their occurrences were strongly associated with a positive air-sea temperature difference, which occurs when warmer air from the southwest flows over the colder waters of the mid-Atlantic, thereby inducing a stable stratification. These events largely occurred in spring and early summer when the air-sea temperature difference was greatest, and very seldom in fall and winter when the air-sea temperature difference is the lowest. The atmospheric conditions leading to these high-shear events is consistent with previous work (Colle and Novak, 2010; Zhang et al., 2006), which had attributed offshore LLJs closer to the coast. The measurements analyzed herein reveal that the high shear and jets persist further from the coast, at offshore distances where wind development is planned.

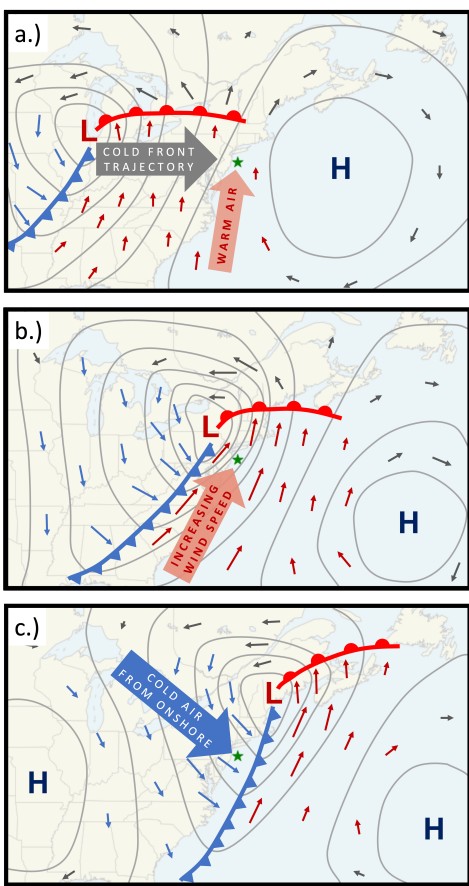

**Figure 14.** A simplified schematic of the synoptic conditions for high-shear events at the beginning (a), during (b), and as the event ends (c). Grey lines represent theoretical isobars, arrows represent typical wind directions, speed, and relative air temperature to the floating lidars (green star), L and H represent low- and high-pressure centers, respectively.

The high-shear events were characterized by low turbulence: $\sim 4.7\%$ TI on average, in contrast to 8.1% when all the data are considered. We note, however, that the accuracy of TI measurements from the floating lidars was not assessed in this study. Future work examining such accuracy would be valuable, provided of course that high-frequency wind speed measurement by the floating lidar is made available.

The LLJ events were especially notable, given their dominant nose heights of 80 m and 100 m and the impact such profiles will have on turbine power generation. Although these events were fairly infrequent, this fact likely has more to do with the upper limit of 200 m from the lidar measurements. Had measurements been available above this height, it is likely that many of the identified monotonic shear events may actually be LLJs with noses above 200 m. Given increasing wind turbine hub heights and rotor diameters (e.g., the IEC 15-MW reference turbine with blade tips extending up to 300 m), further analysis of LLJs above 200 m is warranted.

In identifying these events, we relied on the wind speed gradient, $\Delta U/\Delta z$, rather than the industry standard power law exponent, $\alpha$ (Commission, 2019). The $\alpha$ parameter is nondimensional and does not consider the magnitude of wind speeds.

Consequently, we found that extreme wind shear events could have low values of $\alpha$, while, conversely, low-magnitude wind speed events could have high values of $\alpha$. These results suggest revisiting the standard use of $\alpha$ in turbine design standards and the consideration of alternative parameters such as $\Delta U/\Delta z$.

The public availability of floating lidar data was crucial for this analysis. Although many floating lidars are currently deployed in U.S. offshore wind areas, most data are kept confidential and not available for these types of analyses. Moving

forward, future availability of additional floating lidars will be valuable in further characterizing the regional differences in extreme wind shear events and how they depend on factors such as proximity to the coastline, latitude, and seasonal changes in SST. Furthermore, these floating lidars will become vital in validating NWP models in offshore wind areas, especially their ability to accurately predict these high-shear events.

*Author contributions.* MD led the data analysis with significant contributions from PD and PH. MD wrote the article with equal contributions

from MO, PD, and PH. NB downloaded and processed the lidar data.

*Competing interests.* The authors declare that they have no conflict of interest.

*Acknowledgements.* This work was authored by the National Renewable Energy Laboratory, operated by Alliance for Sustainable Energy, LLC, for the U.S. Department of Energy (DOE) under Contract No. DE-AC36-08GO28308. Funding for this work was provided by the Bureau for Ocean Energy Management under Contract Number IAG-19-02122-1. The views expressed in the article do not necessarily

represent the views of the DOE or the U.S. Government. The U.S. Government retains and the publisher, by accepting the article for publication, acknowledges that the U.S. Government retains a nonexclusive, paid-up, irrevocable, worldwide license to publish or reproduce the published form of this work, or allow others to do so, for U.S. Government purposes. Neither NYSERDA nor OceanTech Services/DNV GL have reviewed the information contained herein and the opinions in this report do not necessarily reflect those of any of these parties.

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
