# Peer review of "Extreme Wind Shear Events in U.S. Offshore Wind Energy Areas and the Role of Induced Stratification"

_Wind Energy Science, 2020_

## Referee Comment (RC1) · Anonymous Referee #1 · 11 Oct 2020

It is a well written manuscript dealing with adverse meteorological conditions of high wind shear off-shore in an wind resource lease area on the east coast of the US. There is considerable interest for adverse meteorological conditions due to the rapid developments of the off-shore wind farms in the US. As such the analysis of the data is very relevant and of considerable interest for the wind energy industry.

From a more general point of view of applied science, I find the discussion on the use of the shear exponent in LLJ and high wind shear conditions very important. The observation in the manuscript, that the shear coefficient (usually denoted alpha) is not a good measure of the extreme wind shear conditions that might pose problems

to wind turbines is very important, because the use of the shear exponent (alpha) is recommend in the IEC (2019) standard. I therefore suggest putting more emphasis on this shortcoming of the shear exponent and even mention the finding in the abstract.

Here are some specific comments:

1. Lines 180 – 190 and table 2. With an uncertainty of 0.1 degree C on the temperature plus any unknown bias in the temperature measurements, it is not reasonable to give the temperature with 3 decimals. Furthermore the difference in temperature between the two sites is within the uncertainty of the observations. This makes these findings scientifically weak, dubious and non-convincing. I suggest simply to remove.

2. Line 62: VLLJ is not defined – actually why introduce VLLJ and not just name it LLJ, which is very well established meteorological phenomena.

3. Line 104: The usual and well established drop off is 2 m/s, please comment on this in the manuscript and explain why this generally accepted value is not applied here.

---

## Referee Comment (RC2) · Anonymous Referee #2 · 11 Oct 2020

General impression

This manuscript describes the occurrence of high wind shear events off the US east coast, based on recent floating LIDAR observations. This is a highly relevant topic for wind energy applications, and since such measurements are quite scarce, this new dataset is very valuable. The analysis is decent, the presentation mostly clear with good quality figures. I like how the authors present their material in a way that is relevant and accessible for engineering applications, without sacrificing too much meteorological rigor. In this review I will focus mostly on the meteorological rigor.

The authors state that their "goal is to characterize these events and understand the

physical mechanisms governing their onset and dissipation". They did a great job in characterizing the events. When it comes to understanding, however, there are some points that require attention:

1. The authors focus exclusively on the US east coast, not only in their results, but also in their attribution of previous work. However, there are many more studies that have thoroughly analyzed similar events in other parts of the world, and provided much insight into their mechanisms. Disregarding these studies not only hampers the stated objective of understanding the phenomena observed in this particular location, but also feels a bit disrespectful. This feeling is reinforced by the notion that the authors coin a new term ("very low-level jets") for something that's been described many times before under the common name "low-level jet".

2. The authors explicitly state that they don't expect diurnal effects in the 'offshore' environment, which suggests that they have not considered the role of coastal mesoscale dynamics at all. By contrast, many of the earlier studies emphasize the role of the coastline in establishing a horizontal temperature contrast between land and sea, which gives rise to a plethora of mesoscale phenomena such as sea breezes, thermal low pressure areas, and baroclinically forced coastal jets. While many of the results presented in this manuscript are consistent with these theories, these theories are not discussed, the hypotheses not falsified. This makes it hard to confide in the overall rationale, even if it is quite sound in many places.

3. I think the manuscript does not accurately depict the role of atmospheric stability. First of all, the metric used to quantify 'atmospheric stability' (the difference between 2m air temperature and sea-surface temperature) is not actually a measure of temperature stratification in the atmosphere. Rather, it is a proxy for the buoyancy forcing at the bottom boundary, which is much more an 'external forcing' than the 'internal parameter' that is atmospheric stability. While the authors seem to be quite careful not to use these terms interchangeably, the presentation in its current form is prone to misinterpretation. Secondly, the authors often refer to stability as the 'driver' or 'cause' of

enhanced wind shear. Moreover, stability, wind shear and turbulence are sometimes discussed as if they are separate quantities, while in fact they are tightly linked: in a stably stratified atmosphere, buoyancy acts to suppress turbulence, while wind shear acts to enhance is. The result is a very delicate balance. If the shear term 'wins', turbulence will be produced, and will act to dissolve both the wind speed and temperature gradients. If the buoyancy term 'wins', turbulence will be suppressed, which 'permits' or 'enables' the vertical gradients to grow. But then the competition starts all over again. So it doesn't really make sense to speak of stability as a cause or driver, because it is very much part of the phenomenon itself.

4. I'm not sure if I understand the jet detection criteria, I think the authors may have made a small mistake here. It would be great if they could share their code, because that would help me to verify the method and reproduce the results.

5. Directional wind shear ('veer') is not discussed. I wonder how much of the 'wind speed shear' discussed herein can be attributed to changes of wind direction with height.

More details to clarify these concerns may be found in my specific comments.

Specific remarks

P2 L38: This statement is too easy. I understand that the authors focus on the US east coast, but when it comes to understanding the low-level jets, I think it would be fair to also acknowledge some studies focusing on different areas as well. For example, Burk and Thompson (1996) and Parish (2000) studied similar events on the US west coast, and they provide a much more complete physical interpretation. In addition to static stability, they argue that the coastal discontinuity introduces a horizontal temperature contrast, and that the associated baroclinity and geostrophic adjustment play an important role in the establishment of the low-level jets. Colle and Novak (2010), which have been cited by the authors, state that they find 'qualitative agreement' with these findings. Furthermore, there are numerous

studies in Europe, e.g. at the Hovsore measurement site in Denmark, the North- and Baltic seas, or the Iberian Peninsula, for example: - https://doi.org/10.1175/1520-0493(1996)124%3C0668:TSLLJA%3E2.0.CO;2 - https://doi.org/10.1175/1520-0450(2000)039%3C2421:FOTSLL%3E2.0.CO;2 - https://doi.org/10.1002/qj.2386 - https://doi.org/10.3402/tellusa.v66.22377 - https://doi.org/10.1002/joc.5303 - https://www.jstor.org/stable/43749611?seq=1 - https://doi.org/10.5194/wes-4-193-2019 - https://doi.org/10.3390/en13143670

P2 L38: It would further increase the relevance of this paper for the audience of WES if it was put in context to the global occurrence of the phenomena. In addition to the references mentioned above, the global climatology of coastal jets in Ranjha (2013) and Lima (2018) might also be a relevant sources. - https://doi.org/10.3402/tellusa.v65i0.20412 - https://doi.org/10.1175/JCLI-D-17-0395.1

P3 L45: It would be good to also discuss the use and limitations of reanalysis data. Especially the most recent datasets with unprecedented resolution have shown at least some skill in reproducing coastal stratification and low-level jets, and they can overcome the spatial limitations of observational studies. See e.g. the global climatology by Lima and the study by Kalverla mentioned above.

P3 L58: While I agree that these datasets are important and perhaps unprecedented, I think the authors are overstating their comprehensiveness and relevance. I don't think it's justified to claim these measurements are representative for all lease areas in figure 1. The coastline in this figure stretches over 8 degrees (!) latitude. Also, the buoys are placed quite for offshore, which is definitely valuable, but the coastal morphology has been shown to significantly impact the structure of the boundary layer closer to the coast. A more nuanced statement would be that they enable a first order characterization of the overall/larger scale situation.

P3 L62: A wind speed maximum at 100 m has been reported in many previous studies under the term low-level jet (as opposed to very-low-level jet). I don't think this event is

unique and/or different from those previous studies. Admittedly, there are also studies that describe low-level jets in the lowest 500 or even 1000m. But if the authors choose to coin this new term, it would be good to further expand on its precise definition. What sets it apart from (some) other studies, and which other studies actually report on the same phenomenon under the conventional name?

P3 L64: This reviewer doesn't understand why many authors are so keen to report power-law exponents in situations where they don't apply. The core assumption for this type of fit is that the wind speed increases monotonically with height. This condition is obviously violated in the case of a low-level jet event. One could argue that the profile can be fitted up to the wind speed maximum, but from the description it appears that the authors used a fixed range between 40 and 160 m. Consequently, a power-law fit would underestimate the wind shear for wind speed maxima below 160 m. Of course, the power-law is deeply embedded in engineering standards and practices, and it is good to communicate in some sort of common vocabulary. But in a scientific text it seems inappropriate to present this metric without any discussion of its shortcomings. PS. I'm happy to see such a discussion near the end of section 2. Perhaps the authors can add a short note here in anticipation of this discussion?

P5 L95: I don't really understand criterion 1. It doesn't necessarily identify jets, right? The highest shear will almost always occur near the surface, also for 'conventional' power-law or log-law profiles, and shear will generally decrease as you go upwards. Even in a low-level jet situation, the 'height of maximum shear' doesn't necessarily coincide with the jet nose. Didn't the authors just mean to refer to the height of the maximum wind speed? That would also help to understand criterion 3, where the (height of) the maximum wind speed are used.

P5 L95: Criterion 2 refers to the 'maximum shear' across the rotor layer, which is confusing. Is it the shear over the (entire) rotor layer, or is it the maximum over the shear computed over smaller height increments, such as currently described for criterion 1?

P6 L105: what if no local minimum is detected? Is the lowest wind speed in measurement range used instead?

P6 L107: I'm happy to see the reference to Baas et al., but I think a bit more discussion on how this algorithm differs (or not) from, and credit for, previous studies is appropriate.

P6 L116: While I agree that the bulk wind shear is more relevant than the power-law exponent, this parameter is also sensitive to the depth of the layer over which it is calculated. Especially in the case of low-level jets. I wonder what would happen when the 'maximum' shear from LLJ criterion 1 or 2 is used instead. Another relevant discussion may be found in https://doi.org/10.5194/wes-4-193-2019.

P7 Fig 4a: Very strong figure! I just wonder what the fitted line is supposed to represent. And what it would look like if the y-axis and color-axis are swapped. This is what I would do intuitively, but perhaps the message is stronger as it is now.

P7 Fig 4b: There seems to be a sort of kink near $x = 0.02$. Could there be any physical explanation for this, and if so, would it make sense to use this as a threshold instead of the 90th percentile?

P7 L127: I wonder if this is actually a significant difference. I would say it's the same order of magnitude. Looking at figure 5 it seems that there are more short-lived events on the SW buoy. Is there also a (significant) difference if the cumulative time is used instead of the absolute number of events?

P8 L140: "influence of local conditions . . . particularly, atmospheric stability" up to L145: "role . . . in driving". While it is true that stability is important (otherwise the wind speed gradient is quickly dissolved by turbulent mixing) and arguably a necessary condition, I would be careful not to overstate its importance as a 'driver'. One could also interpret stability and wind shear as two sides of the same coin, or manifestations of the same process. Whereas a 'driver' is more of an external force such as the advection of warm

air over a cold (sea) surface, or the differential heating over land and sea.

P8 L142: Also in agreement with many of the other previous studies mentioned earlier in this review.

P8 L147: Note that Basu (2018) proposed an elegant method to estimate the Obukhov Length based on wind speed observations only: https://onlinelibrary.wiley.com/doi/full/10.1002/we.2203. It would be interesting to compare the two measures of atmospheric stability. Personally, I'm not a big fan of the Obukhov length for its inherent assumptions, and because delta T is a much more direct observation. But in this case the height difference is quite pronounced, so if you find that L and delta T show similar patterns, that would strengthen the analysis. At least don't state that you can't know it. Also note that, since you're comparing air and sea temperatures, this metric strictly represents the bottom forcing rather than the atmospheric temperature stratification. This is probably more relevant, but it has implications for the subsequent discussion, and is prone to misinterpretation.

P8 L149: Note that a similar reasoning goes for TI and the shear exponent: since it's normalized by the mean wind speed, you don't actually measure the relevant bursts. You could make a figure similar to figure 4a. Wouldn't the standard deviation be a better measure in this case?

P9 L156: Note that parallel to the coastline is also "aligned with the land-sea temperature contrast". Compare to literature about "thermal lows", and literature on baroclinic (low-level) jets (not necessarily offshore).

P9 L159: The hypothesis put forward in this paragraph aligns with the 'Blackadar' model of an inertial oscillation. Note that there is also the 'Holton' mechanism, which is also (maybe even more) consistent with the observations shown so far. These mechanisms are explained e.g. in 2 papers on the great plains LLJ: https://doi.org/10.1175/JAS-D-15-0307.1 and https://doi.org/10.1175/JAS-D-14-0060.1 and references therein. Especially this section would be less speculative if

it was presented in the context of these two mechanisms. Also related to my previous notes about 'causes' and 'drivers'.

P9 L167: These are two very nice example cases. Especially at the end of event 2, it strikes me that the temperature change is so abrupt. I wouldn't expect that if the wind was continuing to blow from the same direction with a very long fetch without any changes in surface properties. Such a change in advected air would only occur with a frontal passage and/or perhaps a change in wind direction (in which case that would be the cause, or driver, of the event). So it might be interesting to show wind direction as well. Furthermore, I'd like to put forward that an abrupt change of temperature at some fixed height can also be caused by turbulent mixing after a prolonged period of growing stratification. While I don't think that's what is happening here, I want to stress that temperature stratification is not really a driver of wind shear, but rather a result of the same process. Another interesting point to think about is the spatial dimension. Is it really the case that only temperature is advected, and stratification of wind and temperature build up as a result? Or is the air that is advected air already stratified, and does it just strengthen a bit over time? And vice versa at the end.

P12 Section 3.4: This section is very short and quite superficial. It highlights some small differences in mostly atmospheric temperature between the two buoys, but doesn't proceed to explore why this might be the case. I also miss a discussion of the role of the distance to shore, which is quite different. I suspect this is the most important difference. If, as I have argued, the land-sea temperature difference plays a role in establishing the stratification, then the distance to shore is one of the key parameters. To make this clearer, the authors might also have a look at some literature on the extent of the sea breeze, which is a closely related phenomenon.

P13 L204: I disagree that the absence of a diurnal signature is expected. The distance to shore is probably not large enough to disregard the effect of mesoscale dynamics related to the presence of the coastline, which are largely driven by the diurnal cycle.

P15 L227: likely due to shear. I don't understand this reasoning. I think the more likely reason is given a few lines later, where the authors state that the low levels of turbulence suggest that there is stable stratification. But it is much more clear when you look at it from a different perspective: since stratification is a prerequisite for the formation of low-level jet events, one could state with near-certainty that stable stratification is present here. Subsequently, it makes sense that there is very little turbulence, because the stratification suppresses turbulent mixing. Stratification, high shear/LLJs, and low turbulence all go hand in hand.

P16 section 4: This is a very interesting section, and it connects very well with the use cases presented in figure 8. I would suggest to move this section to immediately after section 3.3. With respect, the sections 3.4 and 3.5 are much less relevant for understanding the events. I would have liked to have some closure on the mechanisms before continuing with some refinements and impact for wind energy.

P16 L232: 'caused by'. As should be clear by now, I think this statement is not accurate. I would suggest rephrasing it as "associated with".

P16 L235: 'generally consist of'. How did the authors analyze this? When it comes to understanding these events, this synoptic analysis is the most important part of the study, and it would deserve a more extensive explanation.

P16 L235: compare with previous studies on 'thermal lows'.

P18 Fig 14: How did the authors come up with this figure? Is it inspired by a general text book example explaining a low pressure system? Did the authors actually see the fronts in real cases? For if this is more like a thermal low, then I would not expect such fronts. I think both are plausible, but it is important to make clear whether we're looking at real data here, or a visualized hypothesis. Many readers will only remember the figure, and I'm not sure whether that take-home message adequately summarizes the discussion in section 4, which has quite a few loose ends.

P17 L267: Even though synoptic charts are published only 6-hourly, plotting the wind direction itself could provide some clues. Additionally, one could draw pressure maps based on recent reanalysis datasets or other published model data that are available at hourly resolution nowadays (e.g. ERA5).

P17 L272: I would expect that LLJs occur under more moderate conditions, as they require quite a subtle balance between processes. Would it be possible to distinguish between two types of events: one with a strong synoptic forcing, such as the winter storms, where the temperature stratification suppresses (to some extent) the strong mechanical turbulence and consequently one would observe strong but monotonic shear; and another type of event where, in the absence of strong synoptic forcing, the mesoscale/coastal dynamics play a much larger role?

Technical

P1 L10: "when when" P1 L15: "government retains . . . government retains" P3 L55: I'm happy to see a reference to the dataset. I noticed the download page also provides a citation statement (near the bottom). It would be good to add this to the reference entry. P7 L126: Difficult sentence due to . . . and . . . and . . . and P9 Fig 6c: the y-ticks are a bit strange since half event counts don't make sense. P10 Figure 7: "dependency . . . on": I'm not a native speaker, but this term to me suggests some sort of causal relationship, which would at this point be unjustified. "Dependence", according to the dictionary, is just a state of not being independent. But shouldn't this be 'dependence of a and b', or dependence between a and b' (like correlation between. . .)? P13 L186: Perhaps use 'difference' instead of 'change'? 'Change' to me suggests that there's a temporal component involved. P13 L203: 'frequent' instead of 'frequently' P16 L241: low turbulence

---

## Author Comment (AC1) · 3 Mar 2021

**Reviewer 2 Comments and Responses: Debnath et al. wes-2020-103**

In this document, the reviewer's comments are in black and the author's responses are in blue.

Firstly, we are deeply thankful to the reviewer for the extensive and detailed feedback they have provided. Indeed, this is the most feedback any of us authors have received on a journal article and appreciate the time the reviewer has taken. We believe the edits we have made based on the reviewer's comments have greatly improved the quality of the manuscript, and we look forward to the next follow-up.

Our response here to this excellent feedback is based on two main considerations: first, we envision this paper as a preliminary evaluation of atmospheric events based on brand new, novel wind profile data in an important wind energy area where we have been fairly blind to date. Second, this preliminary analysis is based only on these observational data and no modeled data, and therefore we are limited in the breadth of analysis into the causes and mechanisms of these high shear and LLJ events.

Based on these considerations, we envision this article as a first-look at very important atmospheric phenomena occurring in a very important area, which could then be leveraged by subsequent studies to comprehensively study the atmospheric conditions leading to the development of the high shear and LLJ events. Therefore, we have not added substantially new content to this manuscript. Rather, based on the reviewer's feedback, we have focused largely on improving the literature review and addressing LLJs from a global perspective, providing more insight (but not evidence) of what may be causing the observations that we're seeing, and adjusting language throughout.

We hope the reviewer finds our responses and edits to their satisfaction, and look forward to further discussion and finalization of the manuscript.

General comments

1.     The authors focus exclusively on the US east coast, not only in their results, but also in their attribution of previous work. However, there are many more studies that have thoroughly analyzed similar events in other parts of the world, and provided much insight into their mechanisms. Disregarding these studies not only hampers the stated objective of understanding the phenomena observed in this particular location, but also feels a bit disrespectful. This feeling is reinforced by the notion that the authors coin a new term ("very low-level jets") for something that's been described many times before under the common name "low-level jet".

We agree with the reviewer that omitting previous research into LLJs globally was a key omission in this manuscript. We have improved the introduction to better capture the previous research in this subject, and we appreciate all of the references the reviewer has provided.

We understand the reviewer's point and have changed the term "vLLJ" to simply "LLJ" throughout the manuscript. We meant no disrespect in using this term, but simply wanted to distinguish LLJ events below and above 200m heights. All the LLJ events detected in this study are below 200m height, but LLJs are not limited to 200m height.

2.    The authors explicitly state that they don't expect diurnal effects in the 'offshore' environment, which suggests that they have not considered the role of coastal mesoscale dynamics at all. By contrast, many of the earlier studies emphasize the role of the coastline in establishing a horizontal temperature contrast between land and sea, which gives rise to a plethora of mesoscale phenomena such as sea breezes, thermal low pressure areas, and baroclinically forced coastal jets. While many of the results presented in this manuscript are consistent with these theories, these theories are not discussed, the hypotheses not falsified. This makes it hard to confide in the overall rationale, even if it is quite sound in many places.

We believe the reviewer is referring to L204 where we describe "no clear diurnal signature in the LLJs is found, as is expected for the offshore environment where diurnal fluctuations are less pronounced than in land". Here we have adjusted the sentence as: "No clear diurnal signature for these LLJ events can be identified from Figure 11b." We note however, that we present and comment on clear diurnal signatures in the frequency of high shear events in Figure 6(a). In terms of attributing the diurnal signature to specific mesoscale phenomena, we refer to our general response to the reviewer where we describe this study as a preliminary investigation of high shear events based only on offshore lidar observations. Given this, we are unable (and believe it is out of scope) to attribute the events to specific phenomena. However, we have added text where we speculate what phenomena may be causing these events, and emphasize a more detailed analysis will be the subject of future studies using NWP simulations.

3.    I think the manuscript does not accurately depict the role of atmospheric stability. First of all, the metric used to quantify 'atmospheric stability' (the difference between 2m air temperature and sea-surface temperature) is not actually a measure of temperature stratification in the atmosphere. Rather, it is a proxy for the buoyancy forcing at the bottom boundary, which is much more an 'external forcing' than the 'internal parameter' that is atmospheric stability. While the authors seem to be quite careful not to use these terms interchangeably, the presentation in its current form is prone to misinterpretation. Secondly, the authors often refer to stability as the 'driver' or 'cause' of enhanced wind shear. Moreover, stability, wind shear and turbulence are sometimes discussed as if they are separate quantities, while in fact they are tightly linked: in a stably stratified atmosphere, buoyancy acts to suppress turbulence, while wind shear acts to enhance is. The result is a very delicate balance. If the shear term 'wins', turbulence will be produced, and will act to dissolve both the wind speed and temperature gradients. If the buoyancy term 'wins', turbulence will be suppressed, which 'permits' or 'enables' the vertical gradients to grow. But then the competition starts all over again. So, it doesn't really make

sense to speak of stability as a cause or driver, because it is very much part of the phenomenon itself.

We agree with the Reviewer. Lines 145 to 149 have been changed to: "In this section, we intend to investigate the relationship among the high-shear events, atmospheric stability, and turbulence. However, we do not have air temperature measurements at different heights to appropriately characterize the atmospheric stability. Instead, we use the difference between 2-m air temperature and the sea-surface temperature as our best proxy for atmospheric stability. We herein denote this air-sea temperature difference as $\Delta T$. Of course, the air-sea temperature difference is more of an external forcing to the atmosphere, but may provide some indication of atmospheric stability, such as when warm air flows over a colder sea inducing a stable stratification". Further, we have adjusted the sentences to avoid saying that stability is the driver of the wind shear (L140, L158) and have removed the words "atmospheric stability" from L140. The new sentence is: "The presence of strong diurnal and seasonal trends in the number of high-shear events suggest the influence of meteorological conditions". We have changed L158 from "the observations in Section 3.3 suggest the role of induced stable stratification in causing these high-shear events" to "the observations in Section 3.3 suggest a positive correlation between the near surface temperature gradient and these high-shear events".

4.  I'm not sure if I understand the jet detection criteria, I think the authors may have made a small mistake here. It would be great if they could share their code, because that would help me to verify the method and reproduce the results.

We are not clear about which specific criteria are not clear to the reviewer. Any specific point is always helpful. We have shared the jet detection criteria code (function) with the reviewer.

5.  Directional wind shear ('veer') is not discussed. I wonder how much of the 'wind speed shear' discussed herein can be attributed to changes of wind direction with height.

We thank the reviewer for pointing out the missing discussion on wind veer. We have added wind veer plots. Figure 9e provides the relationship between wind veer and air-sea temperature difference. The new sentence in the article is, "Similar to wind speed, the shear exponent (Fig. 9c), the maximum wind shear (Fig. 9c), and wind veer (Fig. 9e) are roughly constant when $\Delta T$ is negative before increasing sharply when the difference becomes positive."

A relationship between wind shear and wind veer has been provided in Fig. 9f. However, we think that any possible cause of wind shear to the wind veer can't be discussed based on the observational data presented in this work. We have added a couple of sentences on the wind veer in Section 3.3: As both wind shear and veer increase with positive $\Delta T$, any possible relationship between the wind shear and wind veer is investigated in Figure 9f. It is observed that the wind veer increases with an increase of wind shear. The upward trend of the wind veer

when the wind shear exponent is negative is caused by a low density of the data. Similarly, we are not confident in the relationship above wind shear exponent 0.4."

Specific comments:

P2 L38: This statement is too easy. I understand that the authors focus on the US east coast, but when it comes to understanding the low-level jets, I think it would be fair to also acknowledge some studies focusing on different areas as well. For example, Burk and Thompson (1996) and Parish (2000) studied similar events on the US west coast, and they provide a much more complete physical interpretation. In addition to static stability, they argue that the coastal discontinuity introduces a horizontal temperature contrast, and that the associated baroclinity and geostrophic adjustment play an important role in the establishment of the low-level jets. Colle and Novak (2010), which have been cited by the authors, state that they find 'qualitative agreement' with these findings. Furthermore, there are numerous studies in Europe, e.g. at the Hovsore measurement site in Denmark, the North- and Baltic seas, or the Iberian Peninsula, for example: - https://doi.org/10.1175/1520-0493(1996)124%3C0668:TSLLJA%3E2.0.CO;2 - https://doi.org/10.1175/1520-0450(2000)039%3C2421:FOTSLL%3E2.0.CO;2 - https://doi.org/10.1002/qj.2386 - https://doi.org/10.3402/tellusa.v66.22377 - https://doi.org/10.1002/joc.5303 - https://www.jstor.org/stable/43749611?seq=1 - https://doi.org/10.5194/wes-4-193- 2019 - https://doi.org/10.3390/en13143670

We thank the reviewer for providing us the information about the different literature relevant to this work. We have mentioned in our general response and response to a comment that we agree with the reviewer that omitting previous research into LLJs globally was a key omission in this manuscript. This work provides a preliminary evaluation of atmospheric events only based on local observational data in an important wind energy area without any numerical model data, thus, the causes and mechanisms of these high shear and LLJ events are left for future work. Upon reviewing different similar works done in different parts of the world, we have added the relevant works done on the LLJ onset and mechanisms in the introduction section. Particularly, paragraph 2 and 3 of the introduction section discuss the LLJ as a global phenomenon which occurs at different locations of the world due to different conditions such as coastal topography, land-sea temperature gradient, frictional decoupling, thermal forcing over sloping terrain, and not limited to the U.S. East Coast.

P2 L38: It would further increase the relevance of this paper for the audience of WES if it was put in context to the global occurrence of the phenomena. In addition to the references mentioned above, the global climatology of coastal jets in Ranjha (2013) and Lima (2018) might also be a relevant sources. - https://doi.org/10.3402/tellusa.v65i0.20412 - https://doi.org/10.1175/JCLI-D-17-0395.1

We agree with the reviewer that framing the results from this study in the context of the global occurrence of high shear events would be beneficial to the wind energy community, however, as we have mentioned in our general response, we feel this analysis is out of scope from the current paper where we are strictly analyzing observations from the two Atlantic buoys. We too believe this will be a valuable contribution to the wind energy community and aim to address this topic in future work.

P3 L45: It would be good to also discuss the use and limitations of reanalysis data. Especially the most recent datasets with unprecedented resolution have shown at least some skill in reproducing coastal stratification and low-level jets, and they can over- come the spatial limitations of observational studies. See e.g. the global climatology by Lima and the study by Kalverla mentioned above.

We are analyzing the high-shear events based on the observational data collected with two buoy-based lidars. We are not doing any numerical modeling in this current study that could provide more information about the mechanisms of the detected high-shear events. Due to this, we think a discussion of reanalysis data is out of scope of this study.

P3 L58: While I agree that these datasets are important and perhaps unprecedented, I think the authors are overstating their comprehensiveness and relevance. I don't think it's justified to claim these measurements are representative for all lease areas in figure 1. The coastline in this figure stretches over 8 degrees (!) latitude. Also, the buoys are placed quite for offshore, which is definitely valuable, but the coastal morphology has been shown to significantly impact the structure of the boundary layer closer to the coast. A more nuanced statement would be that they enable a first order characterization of the overall/larger scale situation.

We agree with the reviewer that coastal morphology can impact the wind characteristics of different locations of the wind lease area differently. However, as this data-set is collected within the wind lease area over a year, the data-set is providing a good representation of the extreme events that the wind lease area might face. We have rephrased the sentence as, "these deployments provide the first publicly available, and relevant observational data set for the analysis of wind characteristics in U.S. East Coast active lease areas and, as such, are of immense value for wind energy research."

P3 L62: A wind speed maximum at 100 m has been reported in many previous studies under the term low-level jet (as opposed to very-low-level jet). I don't think this event is unique and/or different from those previous studies. Admittedly, there are also studies that describe low-level jets in the lowest 500 or even 1000m. But if the authors choose to coin this new term, it would

be good to further expand on its precise definition. What sets it apart from (some) other studies, and which other studies actually report on the same phenomenon under the conventional name?

We understand the reviewer's argument and thank the reviewer for bringing this to our attention. Our intention was not to coin a new term but to differentiate between the LLJ events in this study where the nose is observed (below 200 m) and those where we cannot confirm the existence of a nose due to a limited observational height. With that said, we see that this may be problematic to some readers and have removed the term 'vLLJ' term and replaced it with the usual 'LLJ' term throughout the manuscript.

P3 L64: This reviewer doesn't understand why many authors are so keen to report power-law exponents in situations where they don't apply. The core assumption for this type of fit is that the wind speed increases monotonically with height. This condition is obviously violated in the case of a low-level jet event. One could argue that the profile can be fitted up to the wind speed maximum, but from the description it appears that the authors used a fixed range between 40 and 160 m. Consequently, a power-law fit would underestimate the wind shear for wind speed maxima below 160 m. Of course, the power-law is deeply embedded in engineering standards and practices, and it is good to communicate in some sort of common vocabulary. But in a scientific text it seems inappropriate to present this metric without any discussion of its shortcomings. PS. I'm happy to see such a discussion near the end of section 2. Perhaps the authors can add a short note here in anticipation of this discussion?

We would like to remind the reviewer that this article is not only about LLJs. The high shear events are associated with both monotonic shear and LLJ events. We are reporting the limitations of the power-law exponent in monotonic wind shear too. As the power-law exponent is a well-known parameter and being currently used in the wind industry, we think that reporting the power law exponent is useful and the reported values contain a good reference. Both the wind shear exponent and wind speed gradient are calculated between fixed heights (40 m and 160 m). For the LLJs whose nose height is below 160m, the values for both the wind shear exponent and wind shear will be smaller than if these values were calculated between 40 m and the LLJ nose. We have added the below sentences in Section-2.

"Note that we are using fixed heights (e.g., 40 m to 160 m) to calculate the wind shear exponent and wind speed gradient across the rotor. However, the wind shear exponent and wind speed gradient will be underrepresented across the rotor for the LLJ cases which have wind speed maxima below 160 m height.."

We would also like to point out that we already have the below sentences in Section 2 that describe the limitations of the wind shear exponent.

"A relationship plot (Figure 4a) among wind speed at hub height, wind speed gradient across the rotor, $\Delta U/\Delta z$, and shear exponent, $\alpha$, explains that the shear exponent can be very low even though a turbine faces a high wind speed difference across its diameter. The shear exponent is nondimensional and does not consider the magnitude of wind speed that a turbine actually faces. As a result, data points that would normally be considered as high shear by $\alpha$ often have relatively low wind speeds and would not pose a danger to wind turbines."

P5 L95: I don't really understand criterion 1. It doesn't necessarily identify jets, right? The highest shear will almost always occur near the surface, also for 'conventional' power-law or log-law profiles, and shear will generally decrease as you go upwards. Even in a low-level jet situation, the 'height of maximum shear' doesn't necessarily coincide with the jet nose. Didn't the authors just mean to refer to the height of the maximum wind speed? That would also help to understand criterion 3, where the (height of) the maximum wind speed are used.

We thank the reviewer for the careful check of the detection criteria. We have revised the text in the manuscript to reflect what the algorithm actually does. The text was incorrect as the reviewer pointed out. The adjusted text to reflect the detection criteria follow as:

(i)  the height of maximum wind speed is between the second (40 m) and second-to-last (180 m) measurement height,

$$\leq z\,(U_{\max}) \leq 180;$$

(ii)  the wind speed gradient between the rotor bottom and the nose height ($\frac{\Delta U}{\Delta z}\big|_{\text{nose}}$) is greater than the same pre-specified threshold value used for the monotonic-shear detection,

$$\frac{\Delta U}{\Delta z}\bigg|_{\text{nose}} \geq \frac{\Delta U}{\Delta z}\bigg|_{\text{rotor\_threshold}} \; ; \text{and}$$

(iii)  the wind speed drop off above the jet nose meets minimum requirements in terms of dimensional and dimensionless threshold values,

$$\Delta U_{\text{drop}} \geq 1.5 \text{ m/s and } \frac{\Delta U_{\text{drop}}}{U_{\text{nose}}} \geq 10\%$$

P5 L95: Criterion 2 refers to the 'maximum shear' across the rotor layer, which is confusing. Is it the shear over the (entire) rotor layer, or is it the maximum over the shear computed over smaller height increments, such as currently described for criterion 1?

We thank the reviewer for pointing out that the descriptions of our criteria were not clear. We were using the term "shear" to be consistent with the monotonic shear detection, but that ended up causing more confusion. We have reworded criteria (i) and (ii) to reflect accurately what the code is doing (the code and analysis remain unchanged).

(i) the height of maximum wind speed is between the second (40 m) and second-to-last (180 m) measurement height,

$$\leq z\left(U_{\max}\right) \leq 180;$$

(ii) the wind speed gradient between the rotor bottom and the nose height ($\frac{\Delta U}{\Delta z}\big|_{\text{nose}}$) is greater than the same pre-specified threshold value used for the monotonic-shear detection,

$$\frac{\Delta U}{\Delta z}\bigg|_{\text{nose}} \geq \frac{\Delta U}{\Delta z}\bigg|_{\text{rotor\_threshold}} \quad ; \text{and}$$

(iii) the wind speed drop off above the jet nose meets minimum requirements in terms of dimensional and dimensionless threshold values,

$$\Delta U_{\text{drop}} \geq 1.5 \text{ m/s and } \frac{\Delta U_{\text{drop}}}{U_{\text{nose}}} \geq 10\%$$

P6 L105: what if no local minimum is detected? Is the lowest wind speed in measurement range used instead?

We can see the reviewer's confusion and have added the following sentence to the manuscript to make this point more clear, "If a minimum is not found, a jet nose cannot be identified and the profile is not flagged as a LLJ."

P6 L107: I'm happy to see the reference to Baas et al., but I think a bit more discussion on how this algorithm differs (or not) from, and credit for, previous studies is appropriate.

We thank the reviewer for the suggestion and have added the following text and reference to the manuscript to clear this point, "Depending on the threshold of the wind speed drop, the number of the detected events can vary (Kalverla et al., 2019). For most of the analysis in Kalverla et al. (2019), the threshold used for the wind speed drop is 2 m/s. The enforcement of both dimensional and non dimensional wind speed drop off criteria is based on previous work (Baas et al., 2009) but the threshold values are adjusted in magnitude here because of the limited vertical extent of the measurement data available."

P6 L116: While I agree that the bulk wind shear is more relevant than the power- law exponent, this parameter is also sensitive to the depth of the layer over which it is calculated. Especially in the case of low-level jets. I wonder what would happen when the 'maximum' shear from LLJ criterion 1 or 2 is used instead. Another relevant discussion may be found in https://doi.org/10.5194/wes-4-193-2019. (Peter C. Kalverla 2019).

We understand the reviewer's concern particularly for the LLJ cases. The wind speed gradient across the rotor layer (between fixed height, 40m -160m) has limitations for the LLJ cases of

nose heights below 160m. The wind speed gradient used to detect the high-shear events is not limited to the LLJ events. As shown in the figure below, the wind speed gradient is dependent on the depth of the layer and occurs mostly at 60 m or 80 m height. Due to these lower heights, maximum wind speed gradient does not represent the wind shear across the rotor well for the monotonic high shear events (see the criteria for monotonic events).

As a general parameter used both for monotonic and LLJ events, wind speed gradient is a good parameter to consider. Note that seeing that value of maximum wind speed gradient for the LLJ events, we have used this parameter for meaningful discussions of the LLJ events (Figures 8, 9, and 12).

[Figure]

Figure-1: The heights of the maximum wind speed gradient.

P7 Fig 4a: Very strong figure! I just wonder what the fitted line is supposed to represent. And what it would look like if the y-axis and color-axis are swapped. This is what I would do intuitively, but perhaps the message is stronger as it is now.

We thank the reviewer for the compliment and add the following sentences to the manuscript to explain the purpose of the fitted line.

"The fitted black dash line provides the change of extreme wind shear exponent with wind speeds rather than a constant threshold (e.g., 0.2). It explains that the threshold for the extreme wind shear exponent should decrease with an increase of wind speed to properly consider the wind speed gradient across the rotor diameter."

We have tried the suggestion of swapping the y-axis and color axis, however, we feel that the current approach better displays the relationship we are discussing.

P7 Fig 4b: There seems to be a sort of kink near x = 0.02. Could there be any physical explanation for this, and if so, would it make sense to use this as a threshold instead of the 90th percentile?

The kink is for the SW buoy only. We have further checked the data within the kink (0.034 < $\Delta U/\Delta z$ < 0.036) and have not found anything different than the high shear results explained in this manuscript. The data in the kink occur mostly with positive air-sea temperature difference, southwesterly flow, and low turbulence intensity. The height of the maximum wind speed is mostly 200m suggesting that they are not LLJ events only. The below figure provides a histogram of different important variables within the kink. Our criterion is intersecting the kink and we think it is better to keep the current criteria as a general criterion for both buoys.

[Figure]

Figure-2: The histogram of different variables with 0.034 < $\Delta U/\Delta z$ < 0.036

P7 L127: I wonder if this is actually a significant difference. I would say it's the same order of magnitude. Looking at figure 5 it seems that there are more short-lived events on the SW buoy. Is there also a (significant) difference if the cumulative time is used instead of the absolute number of events?

We agree that the difference is not significant. Both buoys are far from the coast (69 km and 114 km from coast), and it is interesting to investigate (and report) any possible differences between the events at these far distances from the coast. As there are measurements from two buoys, we think it is valuable to report the number of the events detected from both datasets. Similar to the number of events, there is no significant difference in the cumulative time. The cumulative time for the SW and NW buoys are 840 hrs and 848 hrs, respectively.

P8 L140: "influence of local conditions . . . particularly, atmospheric stability" up to L145: "role . . . in driving". While it is true that stability is important (otherwise the wind speed gradient is quickly dissolved by turbulent mixing) and arguably a necessary condition, I would be careful not to overstate its importance as a 'driver'. One could also interpret stability and wind shear as two sides of the same coin, or manifestations of the same process. Whereas a 'driver' is more of an external force such as the advection of warm air over a cold (sea) surface, or the differential heating over land and sea.

We appreciate the point the reviewer is making and have adjusted the sentence on L140 by removing the words "particularly, the atmospheric stability" to address this. Additionally, L145 has been changed to "In this section, we intend to investigate the relationship among the high-shear events, atmospheric stability, and turbulence". We have also added the below sentence in Section 3.3 to address the reviewer's concern:

"Of course, the air-sea temperature difference is more of an external forcing to the atmosphere, but may provide some indication of atmospheric stability, such as when warm air flows over a colder sea inducing a stable stratification."

P8 L142: Also in agreement with many of the other previous studies mentioned earlier in this review.

We thank the reviewer for pointing out an area in which we could tie in other studies and have added more references which are done to show the impact of atmospheric stability on LLJs and wind shear. The adjusted sentence into the article is provided below.

"Indeed, we expect this to be the case that follows the well-established relationships between high wind shear, LLJs, and thermodynamic atmospheric stability established by previous works (Sergeevich and Obukhov, 1954; Stull,1988; Poulos et al., 2002; Wharton and Lundquist, 2012; Blackadar, 1957; Holton, 1967; Burk and Thompson, 1996; Ranjha et al., 2013; Parish, 2000)."

P8 L147: Note that Basu (2018) proposed an elegant method to estimate the Obukhov Length based on wind speed observations only: https://onlinelibrary.wiley.com/doi/full/10.1002/we.2203. It would be interesting to compare the two measures of atmospheric stability. Personally, I'm not

a big fan of the Obukhov length for its inherent assumptions, and because delta T is a much more direct observation. But in this case the height difference is quite pronounced, so if you find that L and delta T show similar patterns, that would strengthen the analysis. At least don't state that you can't know it. Also note that, since you're comparing air and sea temperatures, this metric strictly represents the bottom forcing rather than the atmospheric temperature stratification. This is probably more relevant, but it has implications for the subsequent discussion, and is prone to misinterpretation.

The method proposed by Basu (2018) uses similarity theory based on stability functions to estimate the Obukhov Length. These empirical stability functions are developed based on the observational data collected on land. The surface layer of the marine boundary layer is different from land due to the air-wave interactions and marine phenomena. We have not found any studies that have done any validation of this technique or describe the accuracy of this technique in calculating the Obukhov Length in an offshore environment. Without any further validation of Basu's technique for the offshore wind, we are hesitant to apply this work in these extreme offshore wind shear cases.

P8 L149: Note that a similar reasoning goes for TI and the shear exponent: since it's normalized by the mean wind speed, you don't actually measure the relevant bursts. You could make a figure similar to figure 4a. Wouldn't the standard deviation be a better measure in this case?

The standard deviation could be a good variable to represent the turbulence burst which is not a frequent event. The mean wind speed is also a relevant parameter to the fluctuation of the wind speed. The standard deviation is higher on average for the high shear events, which might suggest more turbulence. But the standard deviation is higher because the high-shear events are operating in high wind speed regimes. In this study, on average, the mean wind speed increases with an increase of positive air-sea temperature difference, and it is intended here to study the normalized wind fluctuation.

P9 L156: Note that parallel to the coastline is also "aligned with the land-sea temperature contrast". Compare to literature about "thermal lows", and literature on baroclinic (low-level) jets (not necessarily offshore).

We have added the below sentences and literature to address the reviewer comment.

"The coastline parallel flow has also been identified in previous works (Colle and Novak, 2010; Winant et al., 1988; Hoinka and Castro, 2003; Soares et al., 2014). Although we can't provide an explanation  of this coastline parallel flow due to the limitations of the measurements used in this study, these previous studies have explained this particular flow direction based on detailed observational and numerical model data. The coastal flows are influenced by the high pressure system over the ocean and a low pressure system inland induced by a sharp contrast between high temperature over land and lower temperature over the sea (Winant et al., 1988; Hoinka

and Castro, 2003; Soares et al., 2014). The coast-parallel flow is then generated by the geostrophic adjustment and deflection due to the Coriolis force (Soares et al., 2014)."

 P9 L159: The hypothesis put forward in this paragraph aligns with the 'Blackadar' model of an inertial oscillation. Note that there is also the 'Holton' mechanism, which is also (maybe even more) consistent with the observations shown so far. These mechanisms are explained e.g. in 2 papers on the great plains LLJ: https://doi.org/10.1175/JAS-D-15-0307.1 and https://doi.org/10.1175/JAS-D-14- 0060.1 and references therein. Especially this section would be less speculative if it was presented in the context of these two mechanisms. Also related to my previous notes about 'causes' and 'drivers'.

Connecting the observations to the Blackadar and Holton mechanisms is an interesting comparison and we thank the reviewer for mentioning this. Based on the wind direction and positive air-sea temperature, we can say: "warmer air coming from the southwest encounters the colder waters of the Mid-Atlantic, causing a positive air-sea temperature difference." It is somewhat speculative to connect the classical studies, which are done based on a wide range of observations, to the limited local measurements used in this study, thus, text and the mentioned articles have been carefully added to the manuscript to connect the observations of this study with the 'Holton' mechanism and 'Blackadar' mechanism. After adding the references, the paragraph now read as,

"The observations in Section 3.3 suggest a positive correlation between the near surface temperature gradient and these high-shear events. Depending on the locations, there are several factors such as topography (Winant et al., 1988), thermal forcing over sloping terrain (Holton 1967) can facilitate the LLJ occurrence. Blackadar (1957) explained that LLJs are inertial oscillations in the wind triggered by the rapid reduction in surface stress (e.g., frictional decoupling) in the boundary layer. It is possible that warmer air coming from the southwest encounters the colder waters off the mid-Atlantic causing a positive air-sea temperature difference. This temperature difference would then induce stable stratification where vertical turbulent exchange from surface winds to those aloft would be reduced and a degree of "decoupling" of winds aloft from the surface would occur. Combined with the long ocean fetch where surface roughness is low, this is likely leading to very low turbulence in the winds aloft at the floating lidars, sufficient to cause high wind shear and allow for the formation of low-level jets."

P9 L167: These are two very nice example cases. Especially at the end of event 2, it strikes me that the temperature change is so abrupt. I wouldn't expect that if the wind was continuing to blow from the same direction with a very long fetch without any changes in surface properties. Such a change in advected air would only occur with a frontal passage and/or perhaps a change in wind direction (in which case that would be the cause, or driver, of the event). So it might be interesting to show wind direction as well. Furthermore, I'd like to put forward that an

abrupt change of temperature at some fixed height can also be caused by turbulent mixing after a prolonged period of growing stratification. While I don't think that's what is happening here, I want to stress that temperature stratification is not really a driver of wind shear, but rather a result of the same process. Another interesting point to think about is the spatial dimension. Is it really the case that only temperature is advected, and stratification of wind and temperature build up as a result? Or is the air that is advected air already stratified, and does it just strengthen a bit over time? And vice versa at the end.

We thank the reviewer for pointing out that the wind direction plot could be interesting to show here. We have added wind direction plots for both cases (Figure 8b). The text added to explain the wind direction plots are provided below.

"Notably, the end of the second high-shear event (e.g., event-02) aligns with the switch back to a negative ΔT value and a sharp change of wind direction. The sharp change in air-sea temperature difference and wind direction suggest the evidence of a frontal passage within this event. The wind direction change in the "event01" is not as sharp as the "event-02" but well-correlated with the change of air-sea temperature difference."

P12 Section 3.4: This section is very short and quite superficial. It highlights some small differences in mostly atmospheric temperature between the two buoys, but doesn't proceed to explore why this might be the case. I also miss a discussion of the role of the distance to shore, which is quite different. I suspect this is the most important difference. If, as I have argued, the land-sea temperature difference plays a role in establishing the stratification, then the distance to shore is one of the key parameters. To make this clearer, the authors might also have a look at some literature on the extent of the sea breeze, which is a closely related phenomenon.

In this section, we have reported the air-sea temperature difference between the two buoys. It is hard to provide any distinct explanations about the coastal impact without any observations of different mechanisms including the land-sea temperature difference. However, we have added some sentences as a motivation for this section.

"The two buoys are located at two different locations of the wind lease areas (Table 1). The data from the two buoys can show the combined impact of the coast on the different wind farms that will be installed at different distances from the coast. The high shear events occur with the southwesterly flow, already described in Section 3.3. The wind farms installed close to the SW buoy will face the southwesterly winds first compared to the wind farms installed close to the NE buoy."

In addition, we have also added the below sentences in this section to relate the impact of coast distance on the buoys.

"The SW and NW buoys are ~69 km and ~114 km far from the coast, respectively. The SW buoy which is closer to the coast faces higher air-sea temperature difference than the NW buoy.

It suggests that the coast has impact on the buoys and the impact varies depending on the distance from the coast."

P13 L204: I disagree that the absence of a diurnal signature is expected. The distance to shore is probably not large enough to disregard the effect of mesoscale dynamics related to the presence of the coastline, which are largely driven by the diurnal cycle.

We agree with the reviewer's comments and have adjusted the sentence in question.
Added text: "No clear diurnal signature for these LLJ events can be identified from Figure 11b."

P15 L227: likely due to shear. I don't understand this reasoning. I think the more likely reason is given a few lines later, where the authors state that the low levels of turbulence suggest that there is stable stratification. But it is much more clear when you look at it from a different perspective: since stratification is a prerequisite for the formation of low-level jet events, one could state with near-certainty that stable stratification is present here. Subsequently, it makes sense that there is very little turbulence, because the stratification suppresses turbulent mixing. Stratification, high shear/LLJs, and low turbulence all go hand in hand.

We agree with the reviewer's comment and have adjusted the text accordingly.
L227 now reads, "This is likely connected to stable atmospheric stratification, which has been found to support LLJ formation and suppress turbulence not only on land but offshore in the U.S. East Coast (Colle and Novak, 2010)."

P16 section 4: This is a very interesting section, and it connects very well with the use cases presented in figure 8. I would suggest to move this section to immediately after section 3.3. With respect, the sections 3.4 and 3.5 are much less relevant for understanding the events. I would have liked to have some closure on the mechanisms before continuing with some refinements and impact for wind energy.

We thank the reviewer for the compliment and understand the interest in a detailed explanation into the mechanisms behind these events. In this study we provide a general synoptic overview that applies to the majority of cases and highlight some of the more interesting synoptic features from specific cases. An in-depth study into the different LLJ formation mechanisms of these cases is out of scope of the current paper and left for future research.

P16 L232: 'caused by'. As should be clear by now, I think this statement is not accurate. I would suggest rephrasing it as "associated with".

We appreciate the suggestion and have incorporated the change from "caused by" to "associated with" into the text. The sentence now reads, "Our analysis to this point has demonstrated the frequency of extreme high-shear events that are associated with stable stratification induced by warmer air from the southwest flowing over colder mid-Atlantic waters."

P16 L235: 'generally consist of'. How did the authors analyze this? When it comes to understanding these events, this synoptic analysis is the most important part of the study, and it would deserve a more extensive explanation.

We appreciate the reviewer's interest in the synoptic analysis in this study. The analysis is performed through the use of the WPC's synoptic map archive to inspect the synoptic setup for each case from start to finish. The schematic is developed as a generalization of the synoptic setup for nearly 75% of the 86 case days. We have added the following sentence into this section to clarify the analysis strategy, "In this section, we examine synoptic charts from NOAA's Weather Prediction Center archive (https://www.wpc.ncep.noaa.gov/) for each case to examine the conditions that lead to the arrival of warmer southwest air.".

P16 L235: compare with previous studies on 'thermal lows'.

This synoptic setup is compared to that of Colle and Novak (2010) on L244 which is one of the reviewer-recommended studies on coastal LLJs in New York. Similar to what is found in that study, a high pressure system is found to be situated offshore to the E of the region of interest and southwesterly flow is apparent. This study attempts to give the readers an overview of the general synoptic conditions for these events and not the specific drivers for each event. We have added text to expand on the possible mechanisms of LLJ formation. However, the explicit mentioning of a "thermal low" is omitted as it would be purely speculative to assert that the low pressure over land is generated from daytime heating. In many of the events, the low pressure system to the west contains associated cold and warm fronts; features that would be absent in a thermal low.
The following text has been added to L245, "... *jet formation* (such as downslope winds from near-shore topography, differential heating over land and sea, sloping marine boundary layers, cold water upwelling, etc.), *the synoptic setups*…"

P18 Fig 14: How did the authors come up with this figure? Is it inspired by a general text book example explaining a low pressure system? Did the authors actually see the fronts in real cases? For if this is more like a thermal low, then I would not expect such fronts. I think both are plausible, but it is important to make clear whether we're looking at real data here, or a visualized hypothesis. Many readers will only remember the figure, and I'm not sure whether that take-home message adequately summarizes the discussion in section 4, which has quite a few loose ends.

We have added text to clear up the confusion here; as mentioned in a previous response, we use WPC's synoptic map archive to generate the schematic as a generalization of the synoptic setup for nearly 75% of the 86 case days. We avoid a composite map as the averaging would cause the different locations and intensities of the high and low pressure systems to become diffuse. We would appreciate more specificity into the "loose ends" the reviewer is referring to in this section so that they can be addressed. As to thermal lows, previous responses have addressed the issue that many of the low pressure systems in these cases contain fronts, which, as mentioned by the reviewer, are not expected with thermal lows. While we do not deny the possibility of thermal lows to generate offshore high shear events, the synoptic charts for the vast majority of cases simply do not support the reviewer's hypothesis that these are overwhelmingly induced by thermal lows.

We have added text in several areas to address the comment:

(L233; mentioned previously) "In this section, we examine synoptic charts from NOAA's Weather Prediction Center archive (https://www.wpc.ncep.noaa.gov/) for each case to examine the conditions that lead to the arrival of warmer southwest air."

(L236) " ...depicted in the schematic shown in Fig. 14a. This schematic is a generalization of the synoptic setup for roughly 75% of the 86 days that registered an event."

(L238) "Due to the differences in location and strength of these pressure systems, a composite schematic was avoided as the averaging would generate a diffuse depiction of the environment."

(L242) Removed the sentence, "Of the 86~days that registered an event, nearly 75\% were observed to have this general synoptic setup" as it would have been repetitive after the additions to L236.

P17 L267: Even though synoptic charts are published only 6-hourly, plotting the wind direction itself could provide some clues. Additionally, one could draw pressure maps based on recent reanalysis datasets or other published model data that are available at hourly resolution nowadays (e.g. ERA5).

We appreciate the suggestion to use reanalysis datasets to analyze the cases, however, we feel this is out of the scope of the present research where we aim to identify these extreme shear cases. We provide a general overview of the most common synoptic setup and leave the use of reanalysis and model data to analyze the mesoscale and microscale forcings for future research.

P17 L272: I would expect that LLJs occur under more moderate conditions, as they require quite a subtle balance between processes. Would it be possible to distinguish between two types of events: one with a strong synoptic forcing, such as the winter storms, where the temperature stratification suppresses (to some extent) the strong mechanical turbulence and consequently one would observe strong but monotonic shear; and another type of event where,

in the absence of strong synoptic forcing, the mesoscale/coastal dynamics play a much larger role?

The speculation of the reviewer is appreciated and well taken. From what we have found, events in which a LLJ nose was observed (below 180 m) typically occurred under similar synoptic conditions as those where a LLJ nose wasn't observed. This could be due to there being no nose (monotonic shear events), or that the nose was simply above 180 m and not detectable via the lidars. Both types of events occurred under both strong synoptic conditions and weak synoptic conditions. Thus, we believe that the main difference is not synoptic, but rather based on the instrumentation as is explained in the text.

Technical comments:

P1 L10: "when when"

We have incorporated the suggestion.

P1 L15: "government retains . . . government retains"

We are using the standard format provided by the legal and communication team.

P3 L55: I'm happy to see a reference to the dataset. I noticed the download page also provides a citation statement (near the bottom). It would be good to add this to the reference entry.

We have incorporated the suggestion. The sentence "Neither NYSERDA nor OceanTech Services/DNVGL have reviewed the information contained herein and the opinions in this report do not necessarily reflect those of any of these parties." has been added in the acknowledgement section too.

P7 L126: Difficult sentence due to ... and ... and ... and

We have changed the sentence to "All the events identified based on the detection criteria are marked as "high shear" events. The events presented in this section include both LLJ and monotonic-shear cases."

P9 Fig 6c: the y-ticks are a bit strange since half event counts don't make sense.

We have adjusted the figure.

P10 Figure 7: "dependency ... on": I'm not a native speaker, but this term to me suggests some

sort of causal relationship, which would at this point be unjustified. "Dependence", according to the dictionary, is just a state of not being independent. But shouldn't this be 'dependence of a and b', or dependence between a and b' (like correlation between. . .)?

We have changed the sentence to "The impact of air-sea temperature difference on wind shear and turbulence intensity."

P13 L186: Perhaps use 'difference' instead of 'change'? 'Change' to me suggests that there's a temporal component involved.

We have incorporated the suggestion

P13 L203: 'frequent' instead of 'frequently' P16 L241: low turbulence

We have incorporated the suggestion.

---

## Author Comment (AC2) · 3 Mar 2021

**Reviewer 1 Comments and Responses: Debnath et al. wes-2020-103**

In this document, the reviewer's comments are in black and the author's responses are in blue.

The authors thank the reviewer for the valuable and useful comments. It is believed that the quality of the manuscript has been improved a lot by the suggestions provided.

General comments:

From a more general point of view of applied science, I find the discussion on the use of the shear exponent in LLJ and high wind shear conditions very important. The observation in the manuscript, that the shear coefficient (usually denoted alpha) is not a good measure of the extreme wind shear conditions that might pose problems to wind turbines is very important, because the use of the shear exponent (alpha) is recommend in the IEC (2019) standard. I therefore suggest putting more emphasis on this shortcoming of the shear exponent and even mention the finding in the abstract.

We thank the reviewer for the kind remarks and have added the below sentences into the abstract.

"In designing a detection algorithm for these events, we find that the typical, non-dimensional power law-based wind shear exponent is insufficient to identify many of these extreme events. Rather, the more simple vertical gradient of wind speed is more suitable."

Specific comments:

1. Lines 180 – 190 and table 2. With an uncertainty of 0.1 degree C on the temperature plus any unknown bias in the temperature measurements, it is not reasonable to give the temperature with 3 decimals. Furthermore, the difference in temperature between the two sites is within the uncertainty of the observations. This makes these findings scientifically weak, dubious and non-convincing. I suggest simply to remove.

We agree with the reviewer and have changed the decimal points for the temperature in the table. The sentence "the difference in temperature between the two sites is within the uncertainty of the observations" has been removed.

2. Line 62: VLLJ is not defined – actually why introduce VLLJ and not just name it LLJ, which is very well-established meteorological phenomena.

We understand the reviewer's point and have changed the term "vLLJ" to simply "LLJ" throughout the manuscript. By introducing this term, we simply wanted to distinguish between the LLJ events in which we observed a jet below 200 m height and those in which there was a jet of nose height above 200 m. However, we see that this could come off as a different phenomenon from the well-established LLJ.

3. Line 104: The usual and well established drop off is 2 m/s, please comment on this in the manuscript and explain why this generally accepted value is not applied here.

We have chosen slightly lower value due to the limited vertical extent of the measurements (i.e., 200m). It is described in the text as (L106-L108): the enforcement of both dimensional and non-dimensional wind speed drop off criteria is based on previous work (Baas et al., 2009) but the threshold values are adjusted in magnitude here due to the limited vertical extent of the measurement data available.

---

## Author Response (AR2)

**Reviewer 1 Comments and Responses: Debnath et al. wes-2020-103**

We thank the reviewer for the valuable comment throughout the review process. We believe that the quality of the article has been improved a lot over the review process. In this document, the reviewer's comments are in black and the author's responses are in blue.

In the revised version of the manuscript all of my comments has been satisfactorily dealt with.
I have a few remarks that easily can be dealt with:

Line 46 – Floors et al 2013 is mentioned twice
Thank you for pointing out the mistake. We have corrected it.

Line 326 something is missing in the sentence

Thank you for finding this; the LaTeX command of "citep" was used instead of "citet." We have fixed this error and the sentence now reads, "[a] similar synoptic environment is found in a case study within Nunalee and Basu (2014) where daily low-level jets formed in coastal New Jersey under an area of high-pressure centered over the mid-Atlantic states."

**Reviewer 2 Comments and Responses: Debnath et al. wes-2020-103**

We thank the reviewer for the valuable comment throughout the review process. We believe that the quality of the article has been improved a lot over the review process. In this document, the reviewer's comments are in black and the author's responses are in blue.

Fig 6, 10, 11, 12. I find these overlaid bar plots rather difficult to read. Red is much more pronounced than blue. The (irrelevant) vertical lines on the bars clutter the figure and distract from the important parts. Perhaps consider changing the plot type to line/scatter?

We agree with the reviewer that red bars are more pronounced than blue. Therefore, we have decreased the thickness of the red bars. We have also tried the scatter and line plots suggested by the Reviewer. After checking the three versions of the figure, we decided to keep the updated bar plot.

P9 L170: "but it may" instead of "but may"?
We thank the reviewer for the suggestion and have changed the part of the sentence to read, "but it may provide some indication of atmospheric stability".

Fig 9f: why did you choose to show the shear exponent on the x-axis, rather than the wind speed gradient itself?

This is a good point. We have checked the plot with wind speed gradient. As the wind direction is calculated with the ratio of wind speed components, wind direction is a non-dimensional parameter of wind speed. But wind speed gradient takes care of the wind speed magnitude and shows a different relationship with wind veer in low and high wind speed conditions. We have added the below sentence to clarify this point.
"Note that, as the wind direction is calculated with the ratio of wind speed components, wind shear exponent is better suited than wind speed gradient to show the relationship between the wind veer and wind shear."

P14 L222: "… reasons for having events being observed" please consider rewriting this sentence.

We thank the reviewer for pointing out the unclarity of the sentence. The sentence now reads, "In this section, we briefly explore potential reasons for having 13% additional events at the SW buoy over the NE buoy."

P14 Section 3.4: The content is fine, but the readability of this paragraph is a bit less pleasant than the rest of the manuscript.

We thank the reviewer for the comment. We have read this section and made minor corrections to improve the readability. Without clear guidance into what the reviewer found unclear, we are not sure we have addressed the concerns.

Fig 11a: maybe add a note that this figure may not be entirely representative for the climatology, since inter-annual variability can be substantial?

We agree with the Reviewer. It is hard to say anything definitive about climatology, however, we are reporting what we are observing based on a year of data. The below sentence is added to the section. "It should be noted that this study uses a year of observational data, but multi-year data would be more useful to investigate the seasonal variability and climatology."

Fig 11b: I find it a bit confusing that a polar plot is used here. So far that's only been used for directions. If the reason is that time is circular, then 11a should also be a polar plot. But I think a normal Cartesian plot is easier to interpret. If you decide keep the polar plot, consider rotating it such that it aligns with an analogue clock.
 We understand the reviewer's concern and have rotated the plot to represent the analogue clock format.

Fig 12a: The legend doesn't correspond with the line types shown in the figure

It does. The legend concisely shows that solid, red is for SW Buoy; dashed, blue for NE buoy; thin line is for monotonic shear; thick line is for LLJ.

Fig 12: VLLJ -> LLJ in legend of a, y-axis of b, and x-axis of c. Also fig 13 legend.

Thank you for catching the typos.

Fig 12b: why is there no KDE fit here?
Because we are looking at six discrete bins.

P16 L266: "monotonic shear profiles not included". Why not? So do I interpret this correctly that the "no-shear" and "LLJ" do not add up to 100%, because the monotonic high shear profiles are excluded? Or is it "LLJ" vs "everything else"? The latter seems to be implied in L264.

It is LLJ vs. no shear as the beginning of the paragraph explains.

P17 L278: "comprise of" is this grammatically correct?
We agree that this sentence used questionable grammar. We have changed the sentence to read "...events generally include a surface low-pressure…"

P18 L301: "Expanding to consider ... it may not be a good characterization of all events". What does "it" refer to? The entire synoptic setup with a low and high pressure system, or the additional strengthening and eastward propagation? If it is the latter, please clarify the sentence. If it is the former, it seems to contradict previous statements about 75% of the events.

We see that this sentence structure is confusing and have rewritten the sentences to now read, "Expanding to consider the 25 longest events (averaging 19 hours in duration) shows that only 12 exhibit this eastward propagation and deepening of the low-pressure system. This implies that while the advancing and strengthening low-pressure system is common in the longest events in this area, it may not be a good characterization of all events including those with a much shorter duration."

P18 L306: "not shown is the wind shift". But this has now been added to the figure, right? Maybe consider labeling the panels from a – h instead of 2 x (a – d). That would make it easier to refer to them.

We thank the reviewer for catching this error. We have adjusted the labels and sentence in question to read, "The wind shift from south-southwesterly to west-northwesterly is also shown (Fig. 8f) as would be expected during a typical cold frontal passage at this location."

P18 L315: I'm not sure what "shortwave troughs" are. Can you add a reference or briefly explain?

We appreciate that shortwave troughs may not be known by readers without meteorology backgrounds and have provided the relevant impacts of shortwave troughs on the near-surface conditions within parenthesis that now reads, "... shortwave troughs (which are relatively small scale synoptic disturbances commonly associated with changes in wind direction near the surface but no, or slight, changes in temperature)."

P19 L324: I wasn't too familiar with the "Mid-Atlantic" region. So this leads to Northerly shore-parallel winds at the offshore lease area? How does that work in terms of stability and warm air advection? It feels like a whole different mechanism. It would be helpful to also see a schematic of this setup. Could it be added in figure 14 (as this figure now only uses half the width of a page)? In my opinion, the distinction between two different 'types' of events (or at least their synoptic configuration) is a very important and new feature of this paper, and adding this setup to figure 14 would highlight that point.

We thank the reviewer for this comment and understand how this could read as confusing if not familiar with this region. We have adjusted the sentences to explain that the high pressure system is centered over the coastal Mid-Atlantic and produces weak synoptic flow (i.e., no, or little, advection) which allows for the diurnal processes to dominate the flow field. We agree that this type of event is important, however, the percentage of events with this setup is relatively small. Further, a case with this synoptic setup has been studied thoroughly in the cited paper by Nunalee and Basu (2014). Because of this, we feel that adding another figure to explain this relatively small subset of cases would be redundant from what is shown in Nunalee and Basu (2014) and distract from the synoptic setup that explains a much larger portion of events. The sentence now reads, "For the event days that did not display the setup

illustrated in Fig. 14 (roughly one quarter of event days), 13% displayed synoptic conditions with a surface high-pressure system over the coastal mid-Atlantic region and offshore lease area. This results in weak synoptic flow over the offshore lease area and conditions greatly subject to diurnal processes. A similar synoptic environment is found in a case study within Nunalee and Basu (2014) where daily low-level jets formed in coastal New Jersey under an area of high-pressure centered just offshore of the mid-Atlantic states.

P21 L368: This study is based on 1 year of measurements. It would be valuable to repeat it with more data. Will this dataset be extended?

We really appreciate the eagerness of the Reviewer for pressing the science needs. But most of the time the outcome and extension of the works do not depend on the science objectives but depend on the funding of the project and relevant commitment.  Based on our commitments, we are happy that we could use a year of data. Hopefully, this article has created a good work to show that the collection and analysis of more offshore data will be needed and useful. We refer this suggestion to a future work.

**Associated Editor Comments and Responses: Debnath et al. wes-2020-103**

In general, the two reviewers are satisfied with the revisions that you have made to the manuscript. The authors should be congratulated for a job well done. However, a few minor issues remain to be corrected. Please consider the corrections requested by referee #2 and the few editorial points by referee #1.

Also, I have an extra comment. In L50-52: "Sloping terrain is also an important driver, where wind speeds closer to the surface accelerate faster than those aloft, producing a LLJ (Holton, 1967; Parish and Oolman, 2010; Shapiro et al., 2016; Du and Rotunno, 2014)." How is this relevant to LLJs offshore? Please explain. The issue of sloping terrain is again used in L190.

We thank the associated editor for this relevant comment. The thermal wind is a great driver of the low-level jets, and sloping terrain facilitates the occurrence of the thermal wind (Holton, 1967; Parish and Oolman, 2010; Shapiro et al., 2016; Du and Rotunno, 2014). In the Southern Great Plains, the gradual west-to-east terrain slope can create a horizontal temperature gradient on a daily basis (Whiteman et al. 1997, Parish et al. 1987) and facilitates the LLJ occurrence. The coastal topography is important for the offshore atmospheric boundary layer and marine phenomena (Beardsley 1987, Burk and Thompson 1996). The sea-level is at the minimum height compared to the shore or coastal topography. Considering the flow direction between the land and sea, there is a natural slope between the land and sea which is relevant to the sloping terrain mechanism described for onshore.

---

## Author Response (AR3)

**Associated Editor Comments and Responses: Debnath et al. wes-2020-103**

I have reviewed your final corrections to the manuscript, and I now find your manuscript suitable for publication in WES. However, I would request a few further editorial corrections listed below.

L 32: The correct citation should be Peña, A., not Pena Diaz... Also, a link to the publication is missing. Please include all links to non-journal citations.
All the references are checked and corrected according to the suggestion.

L31-32: Is there a rationale for the order of the references? Also, further in the introduction.
Thank you for pointing it out. The references are now ordered with their publication years.

Figure 2. Please refer to Figure 1/Table 1 for the location of the mast.
We have added the location of the buoy to the caption of Figure-2.

L 110: Please include units
Units are added.

All units must be written exponentially (e.g. W m–2). Please follow the WES guidelines:
https://www.wind-energy-science.net/submission.html#manuscriptcomposition.
All the units in the text are written with exponential format.

L145. "...which are less than 10 hours but some that extend for more than 2 days." For items other than units of time or measure, use words for cardinal numbers less than 10; use numerals for 10 and above (e.g. three flasks, seven trees, 6 m, 9 d, 10 desks).
Adjusted according to the suggestion.

The caption of Figure 8 is not complete. What year? Also, please refer to Figure 1/Table 1 for the location of the mast.
The caption of Figure 8 has been modified to add more information including year and location of the mast.

Please review your references. Many do not follow the WES guidelines, e.g. 419-421. The publisher is not needed. Others are incomplete: L452.
References are checked and 'publishers' are removed.
For 419-421, the reference has more text which were added according to the suggestion of a Reviewer of this article.

**Chief Editor Comments and Responses: Debnath et al. wes-2020-103**

Chief Editor Decision: Publish subject to technical corrections (04 Jun 2021) by Jakob Mann
Comments to the Author:  Please do the technical corrections suggested by the associate editor. Also regarding the references the "Krogh and Bay" is somehow wrong. Their family names are Mikkelsen and Hasger. Please check it carefully as well.

We have incorporated the suggestions provided by the Associated Editor. Thank you for pointing out the mistake. We have corrected the reference.